# A prototype stochastic parameterisation of regime behaviour in the stably stratified atmospheric boundary layer

Carsten Abraham[1], Amber M. Holdsworth[2], and Adam H. Monahan[1]

[1]University of Victoria, School of Earth and Ocean Sciences, P.O. Box 3065 STN CSC, Victoria, BC V8P 5C2, Canada
[2]Institute of Ocean Sciences, Fisheries and Oceans Canada, 9860 W. Saanich Rd., P.O. Box 6000 Sidney, BC V8L 4B2, Canada

**Correspondence:** Carsten Abraham (abrahamc@uvic.ca)

**Abstract.** Recent research has demonstrated that hidden Markov model (HMM) analysis is an effective tool to classify atmospheric observations of the stably stratified nocturnal boundary layer (SBL) into weakly stable (wSBL) and very stable (vSBL) regimes. Here we consider the development of explicitly stochastic representations of SBL regime dynamics. First, we analyse if HMM-based SBL regime statistics (the occurrence of regime transitions, subsequent transitions after the first, and very persistent nights) can be accurately represented by 'freely-running' stationary Markov chains (FSMC). Our results show that despite the HMM-estimated regime statistics being relatively insensitive to the HMM transition probabilities, these statistics cannot all simultaneously be captured by a FSMC. Furthermore, by construction a FSMC cannot capture the observed non-Markov regime duration distributions. Using the HMM classification of data into wSBL and vSBL regimes, state-dependent transition probabilities conditioned on the bulk Richardson number ($\mathrm{Ri_B}$) or the stratification are investigated. We find that conditioning on stratification produces more robust results than conditioning on $\mathrm{Ri_B}$. A prototype explicitly stochastic parameterisation is developed based on stratification-dependent transition probabilities, in which turbulence pulses (representing intermittent turbulent events) are added during vSBL conditions. Experiments using an idealised single column model demonstrate that such an approach can simulate realistic-looking SBL regime dynamics.

## 1 Introduction

A common classification scheme of the stably stratified atmospheric boundary layer (SBL) distinguishes between two distinct regimes, denoted the weakly and very stable boundary layers (respectively wSBL and vSBL, e.g. Mahrt, 1998a; Acevedo and Fitzjarrald, 2003; Mahrt, 2014; van Hooijdonk et al., 2015; Monahan et al., 2015; Vercauteren and Klein, 2015; Acevedo et al., 2016; Vignon et al., 2017b; Abraham and Monahan, 2019a, b, c, hereafter AM19a, AM19b, and AM19c). In this classification scheme the wSBL is characterised by weak stratification, strong wind and shears which produce sufficient turbulence kinetic energy (TKE) to sustain continuous turbulence and vertical mixing despite the stable stratification (e.g. van de Wiel et al., 2012). The vSBL is characterised by strong stratification, low wind speeds, and weak or intermittent turbulence such that vertical coupling of the atmospheric layers weakens. Very stable boundary layers are also sometimes found to display so-called upside down turbulence, in which TKE is generated aloft by strong shears and then transported downwards. Observational data as well as simulations show that to a good approximation in horizontally homogenous conditions the wSBL conforms to the

classical understanding of turbulence in the atmospheric boundary layer, with turbulence quantities decreasing with height and near-surface profiles which are well-described by Monin-Obukhov similarity theory (MOST; e.g. Sorbjan, 1986; Mahrt, 1998a, b, 2014; Pahlow et al., 2001; Grachev et al., 2005, 2013). In the vSBL, on the other hand, turbulence profiles can decouple from the surface (Banta et al., 2007) and MOST breaks down (e.g. Derbyshire, 1999; Banta et al., 2007; Williams et al., 2013; Mahrt, 2011; Optis et al., 2015). Regime structures and transitions are poorly represented in weather and climate models, due both to coarse resolution (vertical and horizontal) and to an imperfect understanding of the diverse physical processes governing the SBL, particularly with regard to the vSBL to wSBL transitions (e.g Holtslag et al., 2013; Mahrt, 2014). In this study we analyse how well the statistics of SBL regime occupation and regime transitions can be described by a two-regime Markovian system, with the goal of using this information to develop a prototype explicitly stochastic parameterisations of turbulence in the SBL for models of weather and climate.

As transitions between the two SBL regimes are a common feature of SBL dynamics around the globe (AM19b) a representation of the effect of these dynamics in weather and climate models is needed. The regime transitions, however, are associated with a range of different mechanisms. Over land, the wSBL to vSBL transition (which for simplicity we denote the collapse of turbulence even though turbulence does not cease entirely) is normally caused by radiative cooling at the surface increasing the inversion strength and suppressing vertical turbulent fluxes of momentum and heat. This process is relatively well understood and has been examined using conceptual and idealised single column models (van de Wiel et al., 2007, 2017; Holdsworth et al., 2016; Holdsworth and Monahan, 2019; Maroneze et al., 2019, accepted in Q. J. Royal Meteorol. Soc.), or direct numerical simulations of stratified channel flows (Donda et al., 2015; van Hooijdonk et al., 2017) and atmospheric boundary layers (e.g. Flores and Riley, 2011; Ansorge and Mellado, 2014). Radiative cooling leads to very shallow boundary layers which are typically not resolved well in large-scale circulation models. Another mechanism for the wSBL to vSBL transition of particular importance over water is the advection of warm air aloft (AM19c), producing vSBL conditions which are not as shallow as those driven by radiative fluxes.

The vSBL to wSBL transition (which we denote the recovery of turbulence) is less well-understood. Mechanisms by which turbulence recovers include the build-up of shear resulting in instabilities, or an increase in cloud cover weakening the stratification through increasing the downwelling longwave radiation (AM19b). Another potential class of processes initiating these transitions is associated with intermittent turbulent events (e.g. Mahrt, 2014, and references within) which have been found to dominate the turbulent transport in vSBL conditions (Nappo, 1991; Coulter and Doran, 2002; Doran, 2004; Basu et al., 2006; Acevedo et al., 2006; Williams et al., 2013). Intermittent turbulence arises from a range of different phenomena such as breaking gravity waves or solitary waves (Mauritsen and Svensson, 2007; Sun et al., 2012), density currents (Sun et al., 2002), microfronts (Mahrt, 2010), Kelvin-Helmholtz instabilities interacting with the turbulent mixing (Blumen et al., 2001; Newsom and Banta, 2003; Sun et al., 2012), or shear instabilities induced from internal wave propagation (Sun et al., 2004; Zilitinkevich et al., 2008; Sun et al., 2015). It has also been suggested from direct numerical simulations that intermittency can arise as an intrinsic mode of the non-linear equations in the absence of external perturbations of the mean flow (Ansorge and Mellado, 2014). Regardless of which process causes the recovery of turbulence, all phenomena are subgrid-scale in state-of-the-art weather and climate models and are typically not included explicitly through process-based deterministic parameterisations.

Although processes in the SBL have been extensively studied, substantial errors of SBL representation persist in weather and climate models (Dethloff et al., 2001; Gerbig et al., 2008; Bechtold et al., 2008; Medeiros et al., 2011; Kyselý and Plavcová, 2012; Tastula et al., 2012; Sterk et al., 2013; Bosveld et al., 2014; Sterk et al., 2015). Misrepresentation of the SBL includes unrealistic decoupling of the atmosphere from the surface (due to misrepresentations of TKE in the vSBL) resulting in runaway surface cooling (Mahrt, 1998b; Walsh et al., 2008), underestimation of the wind turning with height within the boundary layer (Svensson and Holtslag, 2009), overestimation of the boundary layer height (Bosveld et al., 2014), underestimated low level jet speed (Baas et al., 2009), and underestimation of near-surface wind speed and temperature gradients or their diurnal cycle (Edwards et al., 2011).

Accurate simulations of these near-surface properties is particularly important for global and regional weather forecasts of vertical temperature structures, for instance, which control the formation of fog and frost (Walters et al., 2007; Holtslag et al., 2013). More accurate simulations of the SBL regime behaviour are also important for better representations of surface wind variability and wind extremes (He et al., 2010, 2012; Monahan et al., 2011); simulation and assessment of pollutant dispersal, air quality (Salmond and McKendry, 2005; Tomas et al., 2016), harvesting of wind energy (Storm and Basu, 2010; Zhou and Chow, 2012; Dörenkämper et al., 2015); and agricultural forecasts (Prabha et al., 2011; Holtslag et al., 2013).

Global and regional weather and climate models often use an artificially enhanced surface exchange under stable conditions in order to improve simulations of the large-scale flow (Holtslag et al., 2013). This approach has led to the introduction of long-tailed stability functions and minimum background TKE values not justifiable by observations. In such representations, turbulence is artificially sustained under very stable conditions and the two-regime characteristic of the SBL is suppressed, biasing near surface winds and temperature profiles. Without such parameterisations the nocturnal boundary layers can experience a single turbulence collapse which persists for the entire night. Although the long-tailed stability functions in relatively coarse-resolution models are designed to mimic the gridbox-mean of fluxes over many subgrid-scale wSBL and vSBL patches, with increasing horizontal and vertical resolution more accurate process-based parameterisations are necessary. The occurrence of vSBL to wSBL transitions does not appear to depend deterministically on internal or external state variables (AM19a,b), indicating that parameterisations of the effects of these kinds of transitions in weather and climate models may be required to be explicitly stochastic (e.g. He et al., 2012; Mahrt, 2014). In particular, phenomena such as intermittent turbulence events will likely rely on stochastic parameterisations as their structure and propagation are found to be only weakly-dependent on the mean states (e.g. Rees and Mobbs, 1988; Lang et al., 2018). Stochastic subgrid-scale parameterisations to describe the physically different conditions in the SBL have been proposed to help capture the missing variability in the SBL and improve both climate mean states and forecast ensemble spread (e.g. He et al., 2012; Mahrt, 2014; Nappo et al., 2014; Vercauteren and Klein, 2015). Vercauteren and Klein (2015) propose, for instance, an additional Markovian system to switch between times of strong and weak influences of short-timescale but non-turbulent motions on TKE production in the vSBL, causing regime transitions.

In AM19a,b,c it was demonstrated that hidden Markov model (HMM) analysis of Reynolds-averaged mean states can be used as a tool to analyse the SBL regimes at tower sites in many different settings. Independent of the surface type, the climatological setting, or the complexity of the surrounding topography, two distinct regimes in the state variable spaces of

Reynolds-averaged mean states and turbulence are evident. As the HMM analyses provide climatological (that is, based on long-term statistics) transition probability matrices for a two-regime Markovian system, a natural approach to developing stochastic parameterisations of SBL regime dynamics is to investigate if these can be based on 'freely-running' stationary Markov chains (FSMC) using these transition matrices. The first goal of this study is to determine if climatological Markovian transition probability matrices, which are by construction independent of the state of the SBL, are adequate for simulations of the SBL regime dynamics. While the HMM analyses presented in AM19a assume stationary Markov regime dynamics, statistical analyses of the estimated regime sequences show clear evidence of elevated probability of turbulence collapse close to sunset (AM19b). Furthermore, probability distributions of event durations demonstrate a localised maximum corresponding to a recovery time of on average one to two hours after transitions in which a subsequent transition is unlikely, indicating non-Markov behaviour (AM19b). Because of these non-stationary and non-Markov behaviours, a FSMC will never exactly capture all aspects of SBL regime dynamics. However, it might be sufficient to reproduce most statistics of interest.

In order to investigate the potential of an FSMC-based paramterisation, we first analyse how well they can characterize the HMM-based regime statistics. As part of this analysis we consider the sensitivity of the regime sequence estimated by the HMM to perturbations of the persistence probabilities, allowing for a quantification of what ranges of persistence probabilities accurately describe SBL regime statistics in HMM analyses. By comparing this sensitivity analysis with a sensitivity analysis of regime statistics to varying persistence probabilities in a FSMC we quantify what ranges of persistence probabilities are consistent with both SBL regime statistics derived from an HMM analysis and SBL regime statistics simulated in a FSMC. As we demonstrate that FSMCs cannot simulate all SBL regime statistics of interest, we then consider state-dependent SBL regime transition probabilities. Finally, we develop a pragmatic prototype of an explicitly stochastic parameterisation using the derived state-dependent transition probabilities and present preliminary tests in the idealised single column model (SCM) of Holdsworth and Monahan (2019). The study is organised as follows. First a very short review of the observational data used in the HMM analysis is given in section 2, followed by a brief review of the HMM application to the SBL (section 3). Results of simulating statistics in FSMC are shown in section 4, followed by the description of the prototype state-dependent stochastic parameterisation and test simulations in section 5. Conclusions follow in section 6.

## 2 Data

The observational data used in this study have been discussed in detail in AM19a. We present here a short summary of the data. Observational data sets from nine different research towers measuring standard Reynolds-averaged meteorological state variables with a time resolution of 30 minutes or finer are considered (Table 11). The observational levels of wind speeds and temperatures correspond to the reference state variable sets which are used in the HMM analyses to classify the data into SBL regimes (cf. AM19a, Table 11). Substantial differences among the nine experimental sites exist in terms of their surface conditions, surrounding topography, and their meteorological setting. As a simple classification scheme, we distinguish between land-based, glacial-, and sea-based stations.

The land-based stations can be further clustered into different subsets. Both the Cabauw and Hamburg towers lie in flat, humid, grassland areas, although the Hamburg tower is affected by the large metropolitan area of Hamburg. The Karlsruhe tower is located in the Rhine valley, a rather hilly, forested area. The American sites, Boulder and Los Alamos, are located in relatively arid settings and are strongly affected by the surrounding topography of the Rocky Mountains.

The DomeC observatory, the single glacial-based station, is located in the interior of Antarctica and is influenced by completely different surface conditions including high albedo and low roughness length.

The sea-based stations are the offshore research platforms *Forschungsplattform in Nord- und Ostsee* (FINO), located in the German North and Baltic Seas. These sites are characterized by relatively homogeneous local surroundings and a large surface heat capacity. At the FINO towers nights with statically unstable conditions (defined as nights with two or more unstable

datapoints in a night) are excluded as under these conditions wind speed measurements are unreliable (Westerhellweg and Neumann, 2012). Furthermore, at FINO-1 nights with primary wind directions between 280 and 340 degrees are excluded due to mast interference effects in the data. At the other stations such an exclusion is not necessary as three wind measurements with 120 degree separation are taken at each level.

## 3   Brief summary of the hidden Markov model

We now present a brief overview of the HMM analysis with application to the SBL (Monahan et al., 2015, AM19a). An in-depth description of HMM analysis can be found in Rabiner (1989).

We use the HMM to systematically characterize regime behaviour in the SBL from observed data. The HMM assumes that underlying the observations is an unobserved, or hidden, discrete Markov chain ($\mathbf{X} = \{x_1, x_2, \ldots, x_T\}$). The analysis estimates the regime-dependent parametric probability density distribution (pdf) of the observations (described by the parameter set $\lambda$),

the transition probability matrix $\mathbf{Q}$, and a most likely regime path of the Markov chain (known as the Viterbi Path, VP). We associate the different states of the Markov chain with the SBL regimes (wSBL and vSBL). In our analysis we use observations of the three-dimensional vector consisting of the Reynolds-averaged vertically-averaged mean wind speed, wind speed shear, and stratification to define the HMM input vector $\mathbf{Y}$. A detailed justification of this observational input dataset is presented in AM19a. The HMM estimation algorithm makes use of the following assumptions:

1. Markov assumption: the current regime value $i_t$ at $x_t$ depends exclusively on the previous regime of $x_{t-1}$, so:

$$P(x_t = i_t | x_{t-1} = i_{t-1}, x_{t-2} = i_{t-2}, \ldots, x_0 = i_0) = \mathbf{Q_{i_t i_{t-1}}} \qquad \forall t \quad \text{with} \quad i \in \{0, 1\}, \tag{1}$$

where the dynamics of the SBL are governed by $\mathbf{Q}$ (a $2 \times 2$ matrix corresponding to the wSBL and vSBL, respectively) such that $\sum_{i_t} \mathbf{Q_{i_t i_{t-1}}} = 1$.

2. Independence assumption: conditioned on $\mathbf{X}$, values of $\mathbf{Y}$ are independent and identically distributed variables resulting

in a probability of the observational data sequence of

$$P(\mathbf{Y}, \mathbf{X} | \Lambda) = \pi_i p(\mathbf{y}_0 | x_0 = i_0, \lambda_{i_0}) \prod_{t=1}^{T} \mathbf{Q_{i_t i_{t-1}}} p(\mathbf{y}_t | x_t = i_t, \lambda_{i_t}) \qquad \text{with} \quad i \in \{0, 1\} \tag{2}$$

where $\Lambda = \{\lambda_i, \pi_i, \mathbf{Q}\}_{i \in \{0,1\}}$ is the full set of parameters of the HMM, for which $\{\lambda_i\}_{i \in \{0,1\}}$ is the parameter set describing the regime-dependent pdfs of $\mathbf{y_t}$ (taken to be Gaussian mixture models as in AM19a), and $\pi_i$ is the probability that $x_0$ is in regime $i$ (wSBL or vSBL).

3. Stationarity assumption: this analysis assumes that $\mathbf{Q}$ and $\{\lambda_i\}_{i \in \{0,1\}}$ are time-independent.

The goal of the HMM analysis is to estimate $\Lambda$ from $\mathbf{Y}$. Starting from the probability of the observational time series conditioned on the parameters $P(\mathbf{Y}|\Lambda)$ and applying Bayes theorem to obtain $P(\Lambda|\mathbf{Y})$, the problem reduces to a maximum-likelihood estimation which can be iteratively solved to find local maxima via the expectation maximisation algorithm (Dempster et al., 1979). Having estimated $\Lambda$, the most likely regime sequence (the VP) can be calculated. Regime occupation and transition statistics can then be obtained through analysis of the VP. The estimation of the parameters in the expectation-

maximisation scheme for our analysis is described in detail in Rabiner (1989).

One limitation of the HMM model considered is that it assumes stationary statistics, However, nonstationarities linked to the diurnal cycle and seasonal variability are present in the regime statistics of the SBL (cf. next section, AM19b). Generalizations exist which can account for nonstationarities, such as nonhomogeneous HMMs (Hughes et al., 1999; Fu et al., 2013) which condition transition probabilities on the state of external variables. Other clustering techniques such as the finite-element vari-

15 ational approaches also relax the stationarity assumptions (e.g. Franzke et al., 2009; Horenko, 2010; O'Kane et al., 2013). In particular, the finite-element, bounded-variation, vector autoregressive factor method (FEM-BV-VARX) includes both autoregressive dynamics within each regimes and a modulation of regime dynamics to external drivers. For instance, Vercauteren and Klein (2015) were able to use this model to identify different regimes of interaction between submesoscale motions and the turbulence. However, as our analyses find no clear relationship between external drivers (geostrophic wind and cloud cover)

and transitions between regimes of Reynolds-averaged variables (AM19b), we consider stationary HMM analysis in this study in order to investigate the simplest possible approach to a stochastic parameterisation of turbulence under SBL conditions. Using such a relatively simple parameterisation allows us to determine where additional complexity is warranted and assess how well the dynamics are approximated by stationary Markovian systems.

## 4 SBL regime statistics based on 'freely-running' Markov chains

To be useful as the basis of new parameterisations of turbulent fluxes in the SBL, FSMCs should model SBL regime statistics accurately. The statistics we focus on are the event durations and the probabilities of each of: the occurrence of very persistent nights, of the occurrence of at least one transition within a night, and of multiple transitions within a night. Our reference FSMC models use transition matrices $\mathbf{Q}_{\mathrm{ref}}$ obtained from HMM analyses in AM19a (Table 11). In the HMM analysis, the matrix $\mathbf{Q}$ can be held fixed at prescribed values while other parameters and the VP are estimated. Repeating HMM analyses

using such fixed $\mathbf{Q}$ perturbed from $\mathbf{Q}_{\mathrm{ref}}$, we investigate the sensitivity of the regime statistics of corresponding VPs relative to their reference $\mathrm{VP}_{\mathrm{ref}}$. Because the estimated VP is constrained by the observations, its statistics may differ considerably from a FSMC using the same $\mathbf{Q}$. Evaluating the regime statistics in FSMC for a range of different $\mathbf{Q}$ determines the ranges of

persistence probabilities for which SBL regime statistics of a FSMC match those of $VP_{ref}$. Mathematical expressions used to compute the regime statistics of a FSMC using a given $\mathbf{Q}$ are presented in Appendix A. These calculations require specification of the lengths of the nights. As the tower sites are located in the midlatitudes, we use a range of nighttime durations between 8 and 15 hours. In this section we do not consider the glacial-based station, DomeC in Antarctica. Because the duration of the polar nights is much longer than nights at the other midlatitude stations considered, direct comparisons of regime occupation statistics within individual nights are not meaningful.

## 4.1 Comparison of $VP_{ref}$ and FSMC statistics

For a FSMC (using $\mathbf{Q}_{ref}$ in Eqns. A1 and A2), the frequency of the occurrence of very persistent wSBL nights decreases monotonically with the length of the night (Figure 11). Occurrence probabilities of very persistent wSBL nights from the FSMC match those of $VP_{ref}$ in summertime (nights of duration 8 to 10 hours) but are otherwise underestimated.. For nights lasting longer the FSMC underestimates their occurrence. The increase in occurrence probability in $VP_{ref}$ with increasing night length is consistent with larger synoptic-scale variability and stronger mechanical generation of turbulence in winter, but is not accounted for in a FSMC. The occurrence probabilities of very persistent vSBL nights decrease with increasing length of night in $VP_{ref}$, consistent with an increase in mean pressure gradient force. While the FSMC also shows this behaviour, it systematically underestimates the observed occurrence of very persistent vSBL nights.

In $VP_{ref}$ the probability of at least one wSBL to vSBL or vSBL to wSBL transition occurring within a night shows no systematic dependence on the length of the night across the tower sites (Figure 12). In contrast, the occurrence probability of at least one transition in a FSMC (Eqns. A3 and A4) increases with the length of the night, and is larger than the $VP_{ref}$ at all sites (Figure 12, lower panels). The overestimation of turbulence recovery events by the FSMC is slightly larger than that of turbulence collapse events at land-based stations, while the opposite is true at sea-based stations.

The probabilities of the occurrence of a recovery event subsequent to a turbulence collapse in the FSMC (Eqns. A6 and A8, Figure 13) agree better with those of $VP_{ref}$ than do the probabilities of the overall occurrence of at least one wSBL to vSBL transition (Figure 12). Both $VP_{ref}$ and FSMC occurrence probabilities increase with the length of the night, at about the same rate. At land-based stations the $VP_{ref}$ has fewer subsequent turbulence recovery events than expected from a FSMC, and at sea-based sites more are observed than predicted by a FSMC. Distributions of wSBL to vSBL transitions subsequent to recovery events in a FSMC and the $VP_{ref}$ are generally similar with slightly better agreement in summer than during winter (Figure 13, right panels).

The occurrence of subsequent transition events is related to event durations in the vSBL and wSBL. For both types of events, the duration pdfs display clear maxima between one and two hours after preceding transitions, demonstrating that the occurrence of subsequent transitions most often occurs after some recovery period (Figure 14). No two-regime FSMC can account for these recovery periods because event duration pdfs in the FSMC always decay monotonically (equations A5 and A7 using 12 hour nighttime durations). After the initial recovery period, however, event duration pdfs are generally close in the $VP_{ref}$ and FSMC, resulting in a generally good agreement of mean event durations. The qualitative shape of the event duration pdfs is insensitive to season, although during winter the probabilities of longer events increase (longer nights allow more time

for longer events to occur). Consistently, different nighttime durations in the FSMC change the slope of the exponentially decreasing probability functions (steeper and shallower for respectively shorter and longer nights), however, the substantial differences to pdfs estimated from VP$_{\mathrm{ref}}$ remain.

The results above demonstrate the existence of at least two aspects of the regime statistics which qualitatively cannot be
accounted for by a two-regime FSMC: non-stationary and non-Markov behaviour. While many other regime statistics follow qualitatively the behaviour of a FSMC, quantitative differences between the statistics of VP$_{\mathrm{ref}}$ and FSMCs using $\mathbf{Q}_{\mathrm{ref}}$ are substantial. As values on the diagonal of $\mathbf{Q}_{\mathrm{ref}}$ are close to one (Table 11), the theoretical regime statistics calculated from the FSMC are highly sensitive to these values (cf. Eqns. A1-A8). Therefore, we now investigate the sensitivity of VP to perturbed $\mathbf{Q}$ to determine if biases in the SBL regime statistics of the FSMC can be reduced by modest changes of $\mathbf{Q}$.

**4.2  Sensitivity of the VP to perturbed persistence probabilities**

We consider the sensitivity of the VPs to changes of the persistence probabilities ($\mathbf{Q}_{i_t i_{t-1}} = P(i_{t-1} \rightarrow i_t)$ with $i_t = i_{t-1}$ and $i \in \{\mathrm{wSBL},\mathrm{vSBL}\}$) by perturbing $\mathbf{Q}$ from the reference value, holding it fixed, and repeating the HMM analysis. In order to assess if the perturbed VPs are consistent with VP$_{\mathrm{ref}}$ we consider first the occupation consistency between the two (the fraction of time in which both VPs are in the same regime). As in AM19a, we then assess the consistency of the timing of transitions
(simultaneity of transitions in the reference and perturbed VPs) as well as the representation of very persistent nights. These various metrics are then combined to obtain the total VP consistency. For this part of the analysis, we focus on the Cabauw tower data as we have analysed these data extensively in AM19a. The same qualitative results are found using all tower station data we have considered (not shown).

The estimated VP is robust to substantial changes in $\mathbf{Q}$, with an occupation consistency of more than 90 % obtained for
ranges of wSBL and vSBL persistence probabilities between 0.5 and about 0.9999 (Figure 15). Agreement at the 99 % level is found for persistence probabilities between approximately 0.9 and 0.9999. Accurate representation of the timing of transitions is found for both a broad range of low persistence probabilities and for a small range of persistence probabilities between 0.96 to 0.99. The fact that the accuracy of the transitions is above 99 % if both persistence probabilities are below 0.5 (regime transitions in a single step are more probable than remaining in the regime) is a consequence of the high frequency of modelled
transitions improving the ability to capture individual transitions in VP$_{\mathrm{ref}}$ (at the expense of modelling far too many transition events). Because regime transitions are relatively rare, the physically meaningful range of persistence probabilities corresponds to relatively large values of both. The accuracy of the occurrence of very persistent wSBL nights in the perturbed VP is best for high $P(\mathrm{wSBL} \rightarrow \mathrm{wSBL})$ and is weakly sensitive to $P(\mathrm{vSBL} \rightarrow \mathrm{vSBL})$. This result is not surprising as the high wSBL persistence probability ensures that the majority of very persistent wSBL nights as estimated by VP$_{\mathrm{ref}}$ are captured. This
measure is unaffected by any underestimate of the occurrence of very persistent vSBL nights. Complimentary results are found for the occurrence of very persistent vSBL nights.

Each of the five consistency measures discussed capture distinct aspects of agreement between the reference and perturbed VPs. We define total consistency relative to VP$_{\mathrm{ref}}$ as each of the five described VP consistencies exceed a specific threshold At Cabauw, a 99 % total consistency can be achieved for $P(\mathrm{wSBL} \rightarrow \mathrm{wSBL})$ between approximately 0.97 and 0.99 and

$P(\text{vSBL} \to \text{vSBL})$ between 0.98 and 0.99 (Figure 15, bottom right panel). If only a 95 % total VP consistency is required, $P(\text{wSBL} \to \text{wSBL})$ and $P(\text{vSBL} \to \text{vSBL})$ can range approximately between 0.95 and almost 1.

The sensitivity analysis of the estimated regime occupation sequence to changes in **Q** values reveals that reasonably accurate regime statistics can be obtained over a relatively large range of persistence probabilities. We now turn return to FSMC calculations using the ranges of **Q** where the total VP consistency exceeds 95 % to assess if a common range of persistence probabilities exists where statistics of $\text{VP}_{\text{ref}}$ and FSMC are consistent.

## 4.3 Sensitivity of SBL regime statistics to changing persistence probabilities in a FSMC

As discussed above, calculations of the theoretical values of SBL regime statistics from a FSMC require specifying the duration of the night. To compare statistics from $\text{VP}_{\text{ref}}$ and FSMC we define three night time durations representative of individual seasons (Table 12). The statistics from $\text{VP}_{\text{ref}}$ for the individual towers and seasons are listed in Table 13. To account for sampling uncertainty in the SBL regime statistics as estimated from $\text{VP}_{\text{ref}}$, we consider occurrence probabilities in a 10 % error range ($\pm$ 5 %) around the values from $\text{VP}_{\text{ref}}$.

Similar to what was found at Cabauw, across all land-based stations the perturbed VP is not very sensitive to the values of **Q** and a relatively broad range of persistence probabilities allows for a 95 % total VP consistency in the HMM analyses (Figure 16; grey isolines). The persistence probabilities corresponding to the most likely VPs are reasonably similar across the different stations. In Figure 16 the solid, dashed, and dotted lines respectively correspond to persistence probabilities resulting in FSMC probabilities of at least one transition in a night equal to, 5 % below, and 5 % above the $\text{VP}_{\text{ref}}$ values (wSBL to vSBL in red; vSBL to wSBL in black). The range of persistence probabilities for which the FSMC models the $\text{VP}_{\text{ref}}$ occurrence probabilities of very persistent nights within a 10 % uncertainty band is displayed by a red shaded rectangle with a mark for the exact $\text{VP}_{\text{ref}}$ statistics.

Despite accounting for non-stationarity by considering nights of different lengths separately, in general no ranges of persistence probabilities in any season can be identified for which FSMCs are able to model all SBL regime statistics within our imposed uncertainty range. Only at Cabauw in wintertime and Hamburg in spring or autumn does such a range of persistence probabilities exist.

In order to model only a subset of SBL regime statistics (such as the occurrence of SBL regime transitions or very persistent nights) accurately in a FSMC, the required persistence probability values generally fall outside the region of high total VP consistency between the reference and perturbed VPs. This fact is true for all seasons.

At sea-based stations the range of persistence probabilities that ensures good agreement between the $\text{VP}_{\text{ref}}$ and the perturbed VPs is substantially larger than for land-based stations (not shown). The total VP consistency exceeds 95 % for regime persistence probabilities ranging from approximately 0.92 to 0.99. Nonetheless, similar to land-based stations, no persistence probability range can be identified which allows a FSMC to simulate all SBL regime statistics accurately. Again, to obtain only specific SBL regime statistics, ranges of persistence probabilities are required which generally exceed the values assuring good agreement between the reference and perturbed VPs.

The fact that a FSMC is not able to account for all regime statistics (with or without seasonally varying $\mathbf{Q}$) motivates the consideration of other approaches to the parameterisation of regime dynamics. In particular, the use of state-dependent transition probabilities is supported by the relatively well-understood control of the internal SBL dynamics on wSBL to vSBL transitions (e.g. Acevedo et al., 2019; Maroneze et al., 2019, AM19b, AM19c), including the role of surface energy coupling (van de Wiel et al., 2017; Holdsworth and Monahan, 2019). In the next section we present a prototype of such a parameterisation.

## 5 Stochastic parameterisation for SBL regime dynamics

In this section, we develop a prototype explicitly stochastic parameterisation for numerical weather prediction and climate models and test it using an idealised SCM. We first consider the state dependence of transition probabilities on the basis of $\mathrm{VP_{ref}}$ for the simulation of a two-value (wSBL or vSBL) discrete SBL regime occupation variable ($S$). After having estimated functional forms for these conditional probabilites from fits to data, a paramterisation of episodic enhancement of eddy diffusivity by intermittent turbulence bursts is developed. Finally, the application of this paramterisation in the SCM is presented. We emphasize the fact that the explicitly stochastic parameterisation and its tests presented here are intended to be a proof of concept. A formal validation of model experiments against observational data, including systematic sensitivity analyses of the free parameters and an implementation in a more complex single column model will be the subject of a future study.

### 5.1 State-dependent transition probabilities

The Richardson number (Ri) is often used in parameterisations of stratified boundary layer turbulence, and as such is a natural candidate on which to condition probabilities of transitions between states of $S$. For instance, we expect physically that $P(\mathrm{wSBL} \rightarrow \mathrm{vSBL}|\mathrm{Ri})$ should be small for small Ri, but should increase to virtual certainty for sufficiently large Ri.

Due to their coarse vertical sampling, the Reynolds-averaged observational tower data considered only allow for a characterisation of finite differenced approximations to Ri, defined as the bulk Richardson number ($\mathrm{Ri_B}$):

$$\mathrm{Ri_B} = \frac{g}{\overline{\Theta}}(h - s)\frac{\Theta_h - \Theta_s}{W_h - W_s}, \tag{3}$$

where g is the acceleration due to gravity, $\overline{\Theta}$ the mean background potential temperature, $h$ the height of the upper measurement and $s$ the lower measurement height near the surface, and $\Theta$ and $W$ are respectively the potential temperature and wind speed (with heights indicated by subscripts).

To assess the relationship between $\mathrm{Ri_B}$ and transition probabilities (and in particular the robustness of this relationship across different locations) we investigate composites of its evolution during regime transitions at the various tower sites described in section 2. These composites, centred on the time of transitions and extending 90 minutes before and after, provide information about the average behaviour of $\mathrm{Ri_B}$ across transitions. Such composites do not distinguish differences between individual events which may be important for a detailed physical understanding of a specific transition. Furthermore, composite changes may be less sharp than individual ones, due to variations in transition timing below the averaging scale of the data considered.

Across all land- and glacial-based stations $Ri_B$ measured between each observational height and the surface systematically increases (decreases) during turbulence collapse (recovery) events (Figure 17, columns one and three). At sea-based sites changes in $Ri_B$ are not evident. The weak signal in $Ri_B$ at sea-based stations is likely related to the fact that the lowest observational levels are much higher than at other stations (30 m above mean sea level).

In order to compare across all tower sites we concentrate on $Ri_B$ between about 100 m and 10 m ($Ri_B(100, 10)$) for land-based stations. Because of the very shallow inversion at this location, at DomeC we use $Ri_B$ between 4 m and 1 m (cf. AM19c). The distributions of $Ri_B(100, 10)$ show that not only do the mean and median show a systematic behaviour across regime transitions, but so too does the interquartile range (Figure 17, columns two and four). Consistent with previous results the distributions at sea-based stations across transitions do not change.

The $P(\text{wSBL} \rightarrow \text{vSBL}|Ri_B)$ estimated from using the $\text{VP}_{\text{ref}}$ (binned by $Ri_B$ increments of 0.02) shows low transition probabilities across all tower sites (well below 0.01) for $Ri_B$ smaller than about 0.1 (Figure 18, upper left panel). For $Ri_B$ larger than 0.1, $P(\text{wSBL} \rightarrow \text{vSBL}|Ri_B)$ increases linearly at the land-based and glacial-based station to about 0.6 beyond which wSBL conditions are unsustainable. Consistent with the composites in Figure 17, $P(\text{wSBL} \rightarrow \text{vSBL}|Ri_B)$ at sea-based stations is independent of $Ri_B$.

At land-based stations, $P(\text{vSBL} \rightarrow \text{wSBL}|Ri_B)$ demonstrates that vSBL conditions below $Ri_B$ 0.1 are unsustainable (Figure 18, upper right panel). Above $Ri_B \simeq 0.1$ values of $P(\text{vSBL} \rightarrow \text{wSBL}|Ri_B)$ do not approach zero and are approximately independent of $Ri_B$. However, $P(\text{vSBL} \rightarrow \text{wSBL}|Ri_B)$ exhibits considerable variability with no evident systematic behaviour across stations. If implemented into a parameterisation, the approximately state-independent $P(\text{vSBL} \rightarrow \text{wSBL}|Ri_B)$ would result in turbulence recovery transition statistics decoupled from the flow or stratification profiles. As such, it could not account for the recovery time evident in the observed event duration pdfs. This fact, along with the fact that conditional dependence of wSBL to vSBL transitions is entirely different over land than it is over the ocean, suggests that conditioning the transition probabilities on $Ri_B$ is not appropriate.

As an alternative to conditioning on $Ri_B$, we now consider conditioning transition probabilities on stratification. At all sites except DomeC, we represent the stratification by $\Theta_{100} - \Theta_s$. Due to the very shallow boundary layers at DomeC potential temperature differences between about 4 m and the surface (which demonstrate comparable stratification value changes during transitions) are considered. Although the stratification alone does not describe the full dynamical stability of the flow it is among the state variables which display the largest changes across regime transitions (cf. van de Wiel et al., 2017, and AM19c). Moreover, HMM analyses of the stratification alone have been shown to accurately capture the $\text{VP}_{\text{ref}}$ (cf. AM19a). Across the 90 minutes before and after transitions, not only do the composites of stratification demonstrate clear changes (cf. AM19c) but the entire probability distribution shifts (Figure 19).

The derived transition probabilities conditioned on $\Theta_{100} - \Theta_s$ as estimated from $\text{VP}_{\text{ref}}$ (binned by increments of 0.2 K) demonstrate qualitatively similar behaviour at all stations (Figure 18, second row). In contrast to conditioning on $Ri_B$, conditioning transition probabilities on stratification does not show marked differences between land- and sea-based stations. The $P(\text{wSBL} \rightarrow \text{vSBL}|\Theta_{100} - \Theta_s)$ demonstrates an almost linear increase with increasing stability across all tower sites. The turbulence recovery transition, on the other hand, shows very low $P(\text{vSBL} \rightarrow \text{wSBL}|\Theta_{100} - \Theta_s)$ above about 2-3 K but in-

creases rapidly for weaker inversion strengths. To build a state-dependent parameterisation for $S$ conditioned on stratification, conditional transition probabilities discussed above are fit to functional forms. As the wSBL cannot be sustained for strong inversions nor a vSBL for weak inversions (Figure 18, second row), transition probabilities for such conditions are set to one. The functional forms for the stratification-dependent transition probabilities are

$$
5 \quad P(\text{wSBL} \to \text{vSBL}|\Theta_{100} - \Theta_s) = 
\begin{cases}
\alpha\,(\Theta_{100} - \Theta_s) + \delta & \text{for} \quad \Theta_{100} - \Theta_s < 3K \\
1 & \text{for} \quad \Theta_{100} - \Theta_s \geq 3K
\end{cases}
\tag{4}
$$

and

$$
P(\text{vSBL} \to \text{wSBL}|\Theta_{100} - \Theta_s) = \alpha\,\tanh\left(\frac{\Theta_{100} - \Theta_s - \beta}{\gamma}\right) + \delta.
\tag{5}
$$

The best fit parameter and the RMSE for each station are listed in Table 14; the corresponding best-fit functions are shown in Figure 18 (second row). Those fits are similar enough to each other to allow for an assessment of the mean behaviour through all data for which the parameter sets are listed in Table 14 depicted in Figure 18 (third row, solid black line). Using the median parameter set or a best-fit through all data does not change the parameterisation substantially.

## 5.2 Stochastic forcing in the vSBL regime

As described in the introduction, state-of-the-art planetary boundary layer turbulence parameterisations are generally able to produce radiatively driven turbulence collapse. In contrast, mechanisms to explicitly generate the turbulence recovery are too weak or lacking in standard parameterisations. He et al. (2012) showed that a stochastic process representing the effects of intermittent turbulence events can be implemented as an extra source term in the prognostic TKE budget during vSBL conditions, such that these events episodically drive the vSBL into a turbulence active regime. In their approach, however, the generation of intermittent turbulence bursts did not depend on the state of the boundary layer. Here, we propose to introduce a new local variable, the two-value discrete SBL regime occupation variable $S$, tracking SBL regimes. At each time step the occurrence of a regime transition is determined randomly using the instantaneous state-dependent transition probabilities derived above. When $S$ is in the vSBL additional stochastic forcing is added as a representation of the effect of intermittent turbulence bursts. These enhancements occur with random sizes and at random times, and are similar to a compound Poisson process. This approach can also account for the recovery time in the vSBL event durations.

Many models, including the one we consider, represent turbulence fluxes using first order closure. Here, we represent additional stochastic forcing by increasing the diffusivities of heat and momentum:

$$
K(t,z) = K_{SCM}(t,z) + \sum_k SF_k(t,z),
\tag{6}
$$

where $K$ is the diffusivity for momentum and heat, $K_{SMC}$ the diffusivity as determined by the standard SCM parameterisation (cf. Eqn. B7), and $SF_k$ represents the effects of the $k$-th intermittent turbulence pulse parameterised as follows.

1. At each timestep the probability of the occurrence of an intermittent turbulence pulse is given by $\lambda_{SF}\,\mathrm{d}t$, where $\lambda_{SF}$ is its occurrence rate and $\mathrm{d}t$ the model timestep. If a turbulence pulse is determined to occur at time $t_k$, a random number

$r$ is drawn from a uniform-distribution on $[0, R]$ representing the maximum strength of the burst $SF_k$, occurring at time $t_{w_k} = t_k + \tau_w$.

2. The evolution of $SF_k$ is split into growth and decay phases. The relatively short growth phase is introduced to avoid numerical instabilities, while the decay phase represents the dissipation of the intermittent turbulence pulse. Each $SF_k$ is assumed to have a Gaussian profile in the vertical (which is intended to represent the localisation of the enhanced mixing in the region where the turbulence pulse occurs) given by

$$SF_k(t, z) = s_k(t) \exp\left(-\frac{(z - h_k(t))^2}{2\sigma_k^2(t)}\right). \tag{7}$$

3. The strength $s_k(t)$ increases from $t_k$ until $t_{w_k}$ according to a hyperbolic tangent function. Afterwards an exponential decay is prescribed with an eddy overturning timescale $\tau_e$:

$$s_k(t) = \begin{cases} 0 & \text{for} \quad t < t_k \\ 0.505\, r \tanh\left(\frac{t - 0.5\, \tau_w - t_{w_k}}{0.5\, \tau_w} \tanh^{-1}\left(\frac{99}{101}\right)\right) + 0.505\, r & \text{for} \quad t_k \le t < t_{w_k} \\ r \exp\left(-\frac{t - t_{w_k}}{\tau_e}\right) & \text{for} \quad t \ge t_{w_k} \end{cases} \tag{8}$$

4. We assume the centre of the turbulence pulse, $h_k(t)$, to be initiated aloft (cf. AM19c, Figure 4) and to move exponentially towards the surface during the decay phase:

$$h_k(t) = \begin{cases} h_b & \text{for} \quad t < t_{w_k} \\ (h_b - h_e) \exp\left(-\frac{t - t_{w_k}}{\tau_h}\right) + h_e & \text{for} \quad t \ge t_{w_k} \end{cases}, \tag{9}$$

where $h_b$ and $h_e$ denote the heights of the centre of $SF_k(t, z)$ at the beginning and end of the turbulence pulse respectively, and $\tau_h$ is the vertical migration timescale.

5. The width of $SF_k(t, z)$, $\sigma_k(t)$, is assumed to increase until $t_{w_k}$ according to a hyperbolic tangent function which is introduced to avoid numerical instabilities as for $s_k(t)$. The functional form ensures $\sigma_k(t)$ to grow at the same rate as $s_k(t)$. During its decay $\sigma_k(t)$ widens exponentially (representing the interaction of the turbulent patch with its surrounding) with a typical broadening timescale $\tau_\sigma$:

$$\sigma_k(t) = \begin{cases} \frac{\sigma_w + 1}{2} \tanh\left(\frac{t - 0.5\, \tau_w - t_{w_k}}{0.5\, \tau_w} \tanh^{-1}\left(\frac{\sigma_w - 1}{\sigma_w + 1}\right)\right) + \frac{\sigma_w + 1}{2} & \text{for} \quad t < t_{w_k} \\ (\sigma_w - \sigma_e) \exp\left(-\frac{t - t_{w_k}}{\tau_\sigma}\right) + \sigma_e & \text{for} \quad t \ge t_{w_k} \end{cases}, \tag{10}$$

where $\sigma_w$ and $\sigma_e$ are the widths of the turbulence pulse at respectively the time of its maximal strength and end of its lifecycle.

As indicated by Eqn. 6, the effects of multiple overlapping turbulence bursts are taken to be additive. Thus, we assume no interaction between successive turbulence bursts. Below we test the parameterisation in an idealised SCM. The values for the parameters in the stochastic forcing parameterisation as used in the SCM experiments are listed in Table 15.

### 5.3 SCM experiments with explicitly stochastic parameterisation

The SCM we use to test the parameterisation is described in van Hooijdonk et al. (2017) and Holdsworth and Monahan (2019). The governing equations of the SCM are presented in detail in Appendix B. In this study we consider the upper boundary of the model, at which we impose the boundary condition that the flow is geostrophic with a speed of 6 m s$^{-1}$, to be fixed at 5000 m. The lower boundary of the model domain is determined by the momentum roughness length which is set at $z_0 = 0.001$ m over a dry sand surface with density $\rho_s = 1600$ kg m$^{-3}$, specific heat capacity $c_s = 800$ J kg$^{-1}$ K$^{-1}$ and thermal conductivity $\lambda_s = 0.3$ W m$^{-1}$ K$^{-1}$. Furthermore, we assume clear sky conditions.

The explicitly stochastic parameterisation described above results in SBL transitions that are qualitatively in agreement with observations. An example realisation is presented in Figure 110. In this realisation radiative cooling leads initially to a steady increase in stratification and flow stability. Once the vSBL is established (around simulation hour 2) turbulence pulses occur (none of which are individually sufficient to initiate a vSBL to wSBL transition). These turbulence pulses result in heat fluxes slightly larger than observed but of the right order of magnitude (eg. Doran, 2004). The occurrence of multiple smaller turbulence pulses between simulation hours 6-7.5 slowly erodes the stratification until it is sufficiently weakened that a vSBL to wSBL transition becomes sufficiently likely that such a transition occurs. Consistent with observations the simulated vSBL to wSBL transition lags behind the occurrence of the last turbulence burst (AM19c). After the wSBL is established (about simulation hour 7.5) the stratification begins to increase again and a subsequent turbulence collapse occurs approximately 1.5 hours after the recovery event. This recovery time is very close to the peak in the pdf of the wSBL event duration (cf. Figure 14) providing further evidence that these recovery periods in the wSBL are related to the internal dynamics of the wSBL.

Structures of wind and temperature profiles during vSBL to wSBL transitions resemble those of observations (cf. AM19c). Turbulence pulses lead to warming in the lowest 40 m of the boundary layer as turbulent sensible heat fluxes transport warm air from layers between 50 to 150 m towards the surface (Figure 110, middle panels). Enhanced vertical momentum transport results in the near-surface winds first increasing, and then decreasing (as a result of enhanced surface momentum flux; Figure 110, bottom panels). The relative magnitudes of the initial wind speed increase and subsequent decrease are sensitive to the height and width of $SF_k$ (not shown). As the turbulence pulses decrease the stratification, boundary layer heights increase. These results demonstrate that an explicitly stochastic model with state-dependent transition probabilities and a representation of intermittent turbulence pulses in the vSBL can produce regime transitions that are in qualitative agreement with observations.

## 6 Discussion and Conclusions

Recent studies have demonstrated that hidden Markov model (HMM) analysis is an effective tool to classify the nocturnal boundary layer (SBL) into weakly stable (wSBL) and very stable (vSBL) conditions (Monahan et al., 2015, AM19a, AM19b, AM19c). One goal of this study is to investigate if a two-regime 'freely-running' stationary Markov chain (FSMC, obtained from the HMM analysis) is able to simulate SBL regime statistics with sufficient accuracy to be the foundation of a stochastic parameterisation of SBL regimes. We have assessed the performance of the FSMC (using the most likely transition probabilities from HMM anlyses) relative to the observed regime statistics (the distributions of event durations and the probability

of occurrences of very persistent nights (nights without SBL regime transitions), of any regime transitions, and of multiple subsequent transitions).

The nonstationary occurrence probabilities of very persistent nights as estimated from the HMM analyses cannot be accounted for in a FSMC. The occurrence of regime transitions is slightly overestimated by the FSMC. Transitions subsequent to a preceding ones and the mean event durations in each regime are relatively close to the statistics as estimated with the HMM across all sites and seasons. The recovery time between regime transitions, however, is not explainable by any two-regime FSMC.

By fixing the persistence probability matrix and producing new perturbed HMM regime sequences we have quantified the range of persistence probabilities that are consistent with the most likely HMM regime sequence. At all sites considered, we find that the HMM regime sequence varies only slightly for reasonable variations of transition probabilities.

An analysis of the ranges of persistence probabilities for which a FSMC is consistent with the regime statistics of the HMM analyses indicate that at no tower site considered transition probabilities can be identified which allow a FSMC to match all SBL regime statistics. This result is true even when seasonal non-stationarity is accounted for. The non-Markov behaviour of regime occupation and the fact that aspects of regime transitions such as radiatively-driven turbulence collapse can be simulated by models indicate the need for state-dependent transition probabilities in any explicitly stochastic representation of SBL regime transitions.

With the exception of the sea-based stations state-dependent transition probabilities conditioned on the bulk Richardson number ($Ri_B$) exhibit a systematic state-dependent behaviour for wSBL to vSBL. Transitions probabilities for turbulence recovery events, on the other hand, demonstrate approximately state-independent characteristics with little consistency across sites. The lack of robustness of the conditional transition probabilities and weak dependence of turbulence recovery on $Ri_B$ imply that $Ri_B$ is not an appropriate conditioning variable.

State-dependent transition probabilities conditioned on stratification, however, demonstrate a systematic state-dependent behaviour for both types of transitions across all stations. The wSBL to vSBL transition probabilities conditioned on the stratification increase almost linearly up to a threshold while the vSBL to wSBL transition probabilities show a sigmoidal behaviour.

A prototype of an explicitly stochastic parameterisation is developed based on the following foundations. The explicitly stochastic parameterisation uses a new local variable $S$ tracking the SBL regime (wSBL or vSBL). At each time step, the occurrence of a wSBL to vSBL transition is determined randomly using the instantaneous state-dependent transition probabilities. If $S$ is determined to be in the vSBL, episodes of enhanced turbulent mixing are added.

Experiments in an idealised single column model (SCM) confirm that such an approach provides a reasonable representation of SBL regime dynamics. The occurrence of vSBL to wSBL transitions is related to the occurrence of turbulence bursts and lags their occurrence slightly. The simulated responses of temperatures and wind speeds due to the enhanced heat and momentum fluxes towards the surface are comparable to observations. For both transitions, simulated recovery times are consistent with the observed distributions.

We emphasize the fact that the explicitly stochastic parameterisation presented here is only intended to be a proof of concept. The preliminary results suggest that the parameterisation has the potential to simulate SBL regime dynamics in weather and climate models. The observational information on climatological regime statistics, and event duration distributions (cf. AM19b, AM19c, and the present study) can be used in order to tune the presented explicitly stochastic parameterisation to

generate the appropriate SBL regime variability. Due to the fact that event duration distributions and stratification-dependent transition probabilities are similar between the midlatitude tower sites we believe that the transition process at those stations can be parameterised independent of the local complexity of the surface conditions (such as surface type, topography etc.). Although at DomeC similar stratification-dependent transition probabilities can be obtained, the altitude range used to determine stratification is different than at the other sites, suggesting that a generalised parameterisation has to take additional local state

variables into account. Furthermore, even though a systematic behaviour of transition probabilities conditioned on $\mathrm{Ri_B}$ across the different tower sites is absent, $\mathrm{Ri_B}$ is a coarse approximation to $\mathrm{Ri}$. Analyses of other data sets (with higher spatial and temporal resolution) allowing for better approximations or an estimation of the gradient $\mathrm{Ri}$ or $\mathrm{Ri}$-flux are needed to determine if a systematic behaviour is truly absent.

As our study only considers fixed surface and upper boundary conditions, sensitivity analyses of those in the idealised SCM

as well as different resolutions both in time and space must be assessed against different observational case studies. Due to the fact that the first-order closure requires us to consider the effects of intermittent turbulence events as an enhancement of diffusivities for momentum and heat, we have to impose a rather synthetic space time structure of these enhancements. As intermittent turbulence events are associated with the local enhancement of turbulence kinetic energy (TKE), our parameterisation of episodically occurring turbulence bursts can more naturally be implemented in a prognostic TKE scheme as an additional

TKE source term (e.g He et al., 2012, 2019). Such an approach would allow the model to determine the space time structure of turbulence pulses as well as the interaction of turbulence bursts. In the future, we will implement the parameterisation in a more complex SCM (with and without a prognostic TKE scheme) to obtain a more comprehensive assessment of its use in numerical weather prediction and climate models.

Finally, the parameterisation requires further information regarding horizontal dependence of regime statistics, as it is not

reasonable to expect an entire large-scale weather or climate model grid box to always be in one or the other state. This horizontal dependence will be the subject of a future study. Assessment of the dependence length scales relative to the grid box size will allow the determination of the effects of spatial averaging to the gridbox scale on the probability distribution of SBL quantities.

**Appendix A**

In this appendix, we present the calculations of quantities based on 'freely-running' stationary Markov chains. Note that we introduce in the following equations the notation of $P(i_{t-1} \rightarrow i_t)$ instead of $\mathbf{Q_{i_t i_{t-1}}}$ (cf. equation 2) indicating the regime transition probabilities between two timesteps. Furthermore, we replace the mathematical notation of $i \in \{0, 1\}$ for the regime occupation with the actual terms wSBL and vSBL in order to increase the readability.

### A1 Calculation of very persistent regimes

The occurrence probability of very persistent SBL nights in a stationary Markov chain is calculated using the persistence probabilities of the Markov chain (i.e. $P(\text{wSBL} \to \text{wSBL})$ and $P(\text{vSBL} \to \text{vSBL})$) as follows

$$Pr(\text{wSBL}|n) \quad = \pi_{\text{wSBL}} P(\text{wSBL} \to \text{wSBL})^n, \tag{A1}$$

$$Pr(\text{vSBL}|n) \quad = \pi_{\text{vSBL}} P(\text{vSBL} \to \text{vSBL})^n. \tag{A2}$$

where $\pi_{\text{wSBL}}$ and $\pi_{\text{vSBL}}$ are respectively the initial climatological distributions of being in the wSBL or vSBL and $n$ equals the length of the night in hours multiplied by six (corresponding to a data resolution of 10 min)

### A2 Calculation of at least one particular SBL transition occurrence

The probability of the occurrence of a particular SBL transition in a night of duration $n$ can be expressed in terms of the probability of the absence of any transitions and the probability of single transitions of the complementary transition. In the case of the wSBL to vSBL transition the single complementary transitions start in the vSBL is only allowed a transition to the wSBL. Naturally, the reverse is true for vSBL to wSBL transitions. That way we account for all possibilities that definitely do not have a transition of the considered type.

The probability of the occurrence of turbulence collapse is:

$$Pr((\text{wSBL} \to \text{vSBL}|n) > 0) = 1 - \underbrace{\pi_{\text{wSBL}} P(\text{wSBL} \to \text{wSBL})^n}_{\text{prob. of remaining in the wSBL}}$$

$$- \underbrace{\pi_{\text{vSBL}} P(\text{vSBL} \to \text{vSBL})^n}_{\text{prob. of remaining in the vSBL}} \tag{A3}$$

$$\underbrace{- \sum_{t=0}^{n-1} \pi_{\text{vSBL}} P(\text{vSBL} \to \text{vSBL})^t P(\text{vSBL} \to \text{wSBL}) P(\text{wSBL} \to \text{wSBL})^{n-t-1}}_{\text{prob. of only vSBL to wSBL transitions, remaining in the wSBL afterwards}},$$

Equivalently, the probability of a turbulence recovery (vSBL to wSBL transition) is given by

$$Pr((\text{vSBL} \to \text{wSBL}|n) > 0) = 1 - \underbrace{\pi_{\text{wSBL}} P(\text{wSBL} \to \text{wSBL})^n}_{\text{prob. of remaining in the wSBL}}$$

$$- \underbrace{\pi_{\text{vSBL}} P(\text{vSBL} \to \text{vSBL})^n}_{\text{prob. of remaining in the vSBL}} \tag{A4}$$

$$\underbrace{- \sum_{t=0}^{n-1} \pi_{\text{wSBL}} P(\text{wSBL} \to \text{wSBL})^t P(\text{wSBL} \to \text{vSBL}) P(\text{vSBL} \to \text{vSBL})^{n-t-1}}_{\text{prob. of only wSBL to vSBL transitions, remaining in the vSBL afterwards}}.$$

## A3   Calculation of the probability of subsequent turbulence recovery or collapse event occurrences

The probability that a turbulence recovery event occurs after a turbulence collapse in a night of duration $n$ is equal to the sum of the probabilities of all events that include the occurrence of SBL patterns starting at time $t_1$ in the wSBL, and afterwards showing the sequence $\text{wSBL} \rightarrow \overbrace{\text{vSBL} \rightarrow \ldots \rightarrow \text{vSBL}}^{t\times} \rightarrow \text{wSBL}$ with no further subsequent recovery events, i.e. the SBL remains in the wSBL or have a maximum of one more collapse. The last part of this calculation assures that no double counting of sequences with length $t$ occur as the probability calculation of being in the wSBL at time $t_1$ does not include information of the preceding path. The probability of a certain subsequent recovery pattern of length $t$ can then be calculated as

$$
Pr((\text{wSBL} \rightarrow \overbrace{\text{vSBL} \rightarrow \ldots \rightarrow \text{vSBL}}^{t\times} \rightarrow \text{wSBL}|n) > 0) = \sum_{t_1=0}^{n-t-2} (\pi^T \mathbf{Q}^{t_1})_{\text{wSBL}}
$$

$$
P(\text{wSBL} \rightarrow \text{vSBL})P(\text{vSBL} \rightarrow \text{vSBL})^t P(\text{vSBL} \rightarrow \text{wSBL})\left[ P(\text{wSBL} \rightarrow \text{wSBL})^{n-t-t_1-2} \right. \tag{A5}
$$

$$
\left. + \sum_{t_2=0}^{n-t-t_1-3} P(\text{wSBL} \rightarrow \text{wSBL})^{t_2} P(\text{wSBL} \rightarrow \text{vSBL})P(\text{vSBL} \rightarrow \text{vSBL})^{n-t-t_1-t_2-3} \right],
$$

where $\pi$ is the vector of climatological initial probabilities.

To calculate the overall probability that such a subsequent event occurs is then the summation over all possible $t$:

$$
\sum_t Pr((\text{wSBL} \rightarrow \overbrace{\text{vSBL} \rightarrow \ldots \rightarrow \text{vSBL}}^{t\times} \rightarrow \text{wSBL}|n) > 0) = \sum_{t=0}^{n-2} \sum_{t_1=0}^{n-t-2} (\pi^T \mathbf{Q}^{t_1})_{\text{wSBL}}
$$

$$
P(\text{wSBL} \rightarrow \text{vSBL})P(\text{vSBL} \rightarrow \text{vSBL})^t P(\text{vSBL} \rightarrow \text{wSBL})\left[ P(\text{wSBL} \rightarrow \text{wSBL})^{n-t-t_1-2} \right. \tag{A6}
$$

$$
\left. + \sum_{t_2=0}^{n-t-t_1-3} P(\text{wSBL} \rightarrow \text{wSBL})^{t_2} P(\text{wSBL} \rightarrow \text{vSBL})P(\text{vSBL} \rightarrow \text{vSBL})^{n-t-t_1-t_2-3} \right]
$$

Equivalently, the probabilities of subsequent turbulence collapses after recovery events are

$$
Pr((\text{vSBL} \rightarrow \overbrace{\text{wSBL} \rightarrow \ldots \rightarrow \text{wSBL}}^{t\times} \rightarrow \text{vSBL}|n) > 0) = \sum_{t_1=0}^{n-t-2} (\pi^T \mathbf{Q}^{t_1})_{\text{vSBL}}
$$

$$
P(\text{vSBL} \rightarrow \text{wSBL})P(\text{wSBL} \rightarrow \text{wSBL})^t P(\text{wSBL} \rightarrow \text{vSBL})\left[ P(\text{vSBL} \rightarrow \text{vSBL})^{n-t-t_1-2} \right. \tag{A7}
$$

$$
\left. + \sum_{t_2=0}^{n-t-t_1-3} P(\text{vSBL} \rightarrow \text{vSBL})^{t_2} P(\text{vSBL} \rightarrow \text{wSBL})P(\text{wSBL} \rightarrow \text{wSBL})^{n-t-t_1-t_2-3} \right]
$$

To calculate the overall probability that such a subsequent event occurs is then the summation over all possible $t$:

$$\sum_t Pr((\text{vSBL} \rightarrow \overbrace{\text{wSBL} \rightarrow \ldots \rightarrow \text{wSBL}}^{t\times} \rightarrow \text{vSBL}|n) > 0) = \sum_{t=0}^{n-2} \sum_{t_1=0}^{n-t-2} (\pi^T \mathbf{Q}^{t_1})_{\text{vSBL}}$$

$$P(\text{vSBL} \rightarrow \text{wSBL})P(\text{wSBL} \rightarrow \text{wSBL})^t P(\text{wSBL} \rightarrow \text{vSBL}) \left[ P(\text{vSBL} \rightarrow \text{vSBL})^{n-t-t_1-2} \right. \tag{A8}$$

$$\left. + \sum_{t_2=0}^{n-t-t_1-3} P(\text{vSBL} \rightarrow \text{vSBL})^{t_2} P(\text{vSBL} \rightarrow \text{wSBL})P(\text{wSBL} \rightarrow \text{wSBL})^{n-t-t_1-t_2-3} \right]$$

## Appendix B

The idealized SCM is a model of pressure-driven flow in the dry SBL assuming horizontal homogeneity, described in detail in
Holdsworth and Monahan (2019). The model equations follow those of Blackadar (1979):

$$\frac{\partial U}{\partial t} = \frac{1}{\rho}\frac{\partial \tau_x}{\partial z} - \frac{1}{\rho}\frac{\partial p}{\partial x} + f_0 V \tag{B1}$$

$$\frac{\partial V}{\partial t} = \frac{1}{\rho}\frac{\partial \tau_y}{\partial z} - \frac{1}{\rho}\frac{\partial p}{\partial y} - f_0 U \tag{B2}$$

$$\frac{\partial T}{\partial t} = -\frac{1}{\rho c_p}\frac{\partial H}{\partial z} - C_{HL} \tag{B3}$$

$$\frac{\partial T_s}{\partial t} = C_1(I_{lw} - \sigma T_s^4 - H_0) - C_2(T_s - T_d) \tag{B4}$$

where the three state variables $U(z,t)$, $V(z,t)$ and $T(z,t)$ are the zonal velocity, meridional velocity, and potential temperature.
The surface temperature $T_s$ is determined by the sum of radiative, turbulent sensible heat, and surface heat fluxes as described
in more detail below. The constant $C_{HL} = 2$ K h$^{-1}$ represents the atmospheric cooling due to net long-wave radiative flux
divergence and is set as a fixed constant for simplicity.

The geostrophic wind components are defined by

$$U_g = -\frac{1}{f_0 \rho}\frac{\partial p}{\partial y}, \tag{B5}$$

$$V_g = \frac{1}{f_0 \rho}\frac{\partial p}{\partial x}, \tag{B6}$$

with the magnitude of the geostrophic wind speed given by $S_g = (U_g^2 + V_g^2)^{0.5}$.

The vertical heat flux $H = \rho c_p \overline{w'T'}$ and shear stresses $\tau_x = -\rho \overline{U'w'}$ and $\tau_y = -\rho \overline{V'w'}$ (where $w$ is the vertical velocity)
are parameterized using first order closure $\tau_x/\rho = K_m \partial_z U$, $\tau_y/\rho = K_m \partial_z V$ and $H/\rho c_p = -K_H \partial_z T$, where $K_m$ and $K_h$ are
the diffusivities for respectively momentum and heat. The diffusivities are taken to be the sum of molecular and turbulent
contributions (Moene et al., 2010):

$$K_m = l^2 |\partial_z U| f_m(\text{Ri}) + \nu \tag{B7}$$

$$K_h = l^2 |\partial_z U| f_h(\text{Ri}) + \lambda \tag{B8}$$

where the molecular contribution $\nu = 1.5 \times 10^{-5}\,\mathrm{m^2\,s^{-1}}$ is the kinematic viscosity. The molecular Prandtl number is fixed at $\mathrm{Pr} = 0.72$ and $\lambda = \nu/\mathrm{Pr}$ is the molecular diffusivity. The mixing length is given by

$$l = \left(1 - \exp\left(-\frac{u_{0*}z}{C\nu}\right)\right)\left(\frac{\kappa(z - z_0)}{(1 + \kappa(z - z_0)/\lambda_0)}\right) \tag{B9}$$

with $\kappa$ the von Kármán constant, $u_*$ friction velocity, $C = 26$ (Van Driest, 1951), and $\lambda_0 = 0.00027 S_g/f_0$ (Blackadar, 1962).

The stability functions $f_{m,h}(\mathrm{Ri})$, depend on the Richardson number $\mathrm{Ri} = \frac{g}{T_{\mathrm{REF}}}\frac{\partial_z T}{(\partial_z U)^2}$ which are related to the similarity functions from MOST $\phi_{m,h}(\zeta)$ by

$$f_m(\mathrm{Ri}_{eq}) = \phi_m^{-2}(\zeta) \tag{B10}$$
$$f_h(\mathrm{Ri}_{eq}) = \phi_m^{-1}(\zeta)\phi_h^{-1}(\zeta)$$

where $\zeta = z/L$ is the stability parameter and $L = -\dfrac{u_*^2}{\kappa\frac{g}{T_s}\frac{H_0}{c_p\rho}}$ is the Obukhov length. In our simulations, we use the Businger-

10 Dyer formulation given by $\phi_{m,h}(\zeta) = 1 + \beta\zeta$ where $\beta = 1/\mathrm{Ri}_c = 5.2$ (Businger, 1988).

     At the upper boundary of the model we impose the boundary condition that the flow is geostrophic and a no-flux condition so $H = 0$ and $\tau = 0$. The lower boundary of the model domain is determined by the roughness length (assumed to be the same for momentum and energy) with no-slip boundary conditions $U(z_0) = 0$ and $T(z_0, t) = T_s(t)$.

     The model implements the surface energy scheme of Blackadar (1976), known as the force-restore method. The surface,
represented as an infinitesimally thin layer with temperature $T_s(t)$ at $z = z_0$, is forced by the net radiation and sensible heat flux and restored to the subsurface temperature through the subsurface energy fluxes. The damping depth of the diurnal forcing $d = (2\lambda_s/C_s\omega)^{0.5}$, where $C_s = \rho_s c_s$ is the volumetric heat capacity, is associated with a sinusoidal diurnal forcing. The temperature at this depth is set as the subsurface temperature $T_d = 281$ K. In Eq. (B4), $C_1 = 2/(0.95 C_s d)$ and $C_2 = 1.18(2\pi/T_d)$. The first two terms in Eq. (B4) constitute the net long-wave radiation $Q_n$, the third term is the sensible heat flux into the atmosphere
due to turbulent transports $H_0$, and the fourth term is the heat flux into the subsurface $G$. As our focus is on the stably stratified boundary layer we do not include the effects of albedo or latent heat in the heat budget. We also neglect the effects of the vegetation canopy.

     The downwelling longwave radiation is given by

$$I_{lw} = \sigma(Q_c + 0.67(1 - Q_c)(1670 Q_a)^{0.08})T_a^4 \tag{B11}$$

where $Q_c$ is the cloud fraction, $Q_a$ is the specific humidity and $T_a$ is the atmospheric temperature at a reference level $z_a$ just above the Earth surface (Staley and Jurica, 1972; Deardorff, 1978). For simplicity, $Q_a$ is held constant at $0.003\,\mathrm{kg\,kg^{-1}}$.

     The equations are integrated in time using a fourth order Runge-Kutte method. The spatial discretization is obtained using finite differences on a logarithmic grid. This grid has 50 vertical levels with a much finer resolution in the boundary layer than aloft and is determined by $z_j = \Delta z_0 \frac{r^j - 1}{r - 1}$ with a stretch factor $r = \frac{\Delta z_j}{\Delta z_{j-1}} \simeq 1.10$ and an initial step size of $\Delta z_0 = 2$ m. The
30 prognostic variables $U$, $V$, and $T$ are defined at the $z_i$ grid levels, while the diagnostic variables of $H$, $\tau$, and Ri are defined on $z_{i+\frac{1}{2}}$ levels.

We define $t = 0$ as the time when the shortwave radiation goes to zero acknowledging the fact that observations indicate that the onset of the SBL can occur before this time (van Hooijdonk et al., 2017; van de Wiel et al., 2017, AM19a). The initial conditions were set in accordance with the logarithmic equations that arise from MOST. The near-neutral profiles for temperature and wind used to initialize the model are given by

$$
\begin{aligned}
U_0 &= \frac{U_{\text{ext}}}{\kappa} \ln(z/z_0) \\
V_0 &= \frac{V_{\text{ext}}}{\kappa} \ln(z/z_0) \\
T_0 &= T_s + \frac{\theta_{\text{ext}}}{\kappa} \ln(z/z_0)
\end{aligned}
\tag{B12}
$$

where $U_{\text{ext}} = U_g \kappa / \ln(h/z_0)$, $V_{\text{ext}} = V_g \kappa / \ln(h/z_0)$ and $\theta_{\text{ext}} = 0.01\,\text{K}$ (Monin and Obukhov, 1954). For simplicity, we set $\frac{\partial p}{\partial y} = 0$ in all of our simulations, so $U_0$ is identically zero at the start of the simulation.

*Competing interests.* The authors declare that they have no conflict of interest.

*Acknowledgements.* We would like to thank a number of individuals and institutes for their willingness to share their tower data which were indispensable in carrying out this extensive comparison of SBL structures at different location sites. Our acknowledgements are presented in the order that the tower stations were presented in the paper but we are equally thankful to all. The NOAA Earth System Research Laboratory's (ESRL) Physical Sciences Division (PSD) operates the Boulder Atmospheric Observatory (BAO) tower and makes the data publicly available. Information how to obtain the data is given on https://www.esrl.noaa.gov/psd/technology/bao/site/. The Royal Dutch Meteorological Institute (KNMI) is thanked for providing tower data from the Cabauw Experimental Site for Atmospheric Research (CESAR) which can be downloaded at http://www.cesar-database.nl. Felix Ament and Ingo Lange provided data from the Wettermast Hamburg of the Meteorological Institute of the University of Hamburg. Martin Kohler and the Institute for Meteorology and Climate Research of the Karlsruhe Institute of Technology (KIT) provided observations from the turbulence and meteorological mast in Karlsruhe. The team of the Los Alamos National Laboratory (LANL) are thanked making data from the Environmental Monitoring Plan (EMP) freely available which can be downloaded from http://environweb.lanl.gov/weathermachine/data_request_green_weather.asp. The French and Italian polar institutes (IPEV and PANRA, respectively) which operate the DomeC observatory in Antarctica are acknowledged for providing data through IPEV (program CALVA 1013), INSU/LEFE (GABLS4 and DEPHY2), and OSUG (GLACIOCLIM). The data are available on the CALVA website http://lgge.osug.fr/~genthon/calva/home.shtml. The Bundesamt für Seeschifffahrt und Hydrographie (BSH), the Bundesministeriums für Wirtschaft und Energie (BMWi), the Projektträger Jülich (PTJ), and Olaf Outzen are thanked for granting access to the data from the offshore research platforms FINO-1, FINO-2, and FINO-3 in Germany.

Carsten Abraham and Adam H. Monahan are supported by the Natural Sciences and Engineering Research Council Canada (NSERC). Amber M. Holdsworth acknowledges support by the DFO ACCASP program.

Finally, we thank three anonymous reviewers whose suggestions substantially improved this manuscript.

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

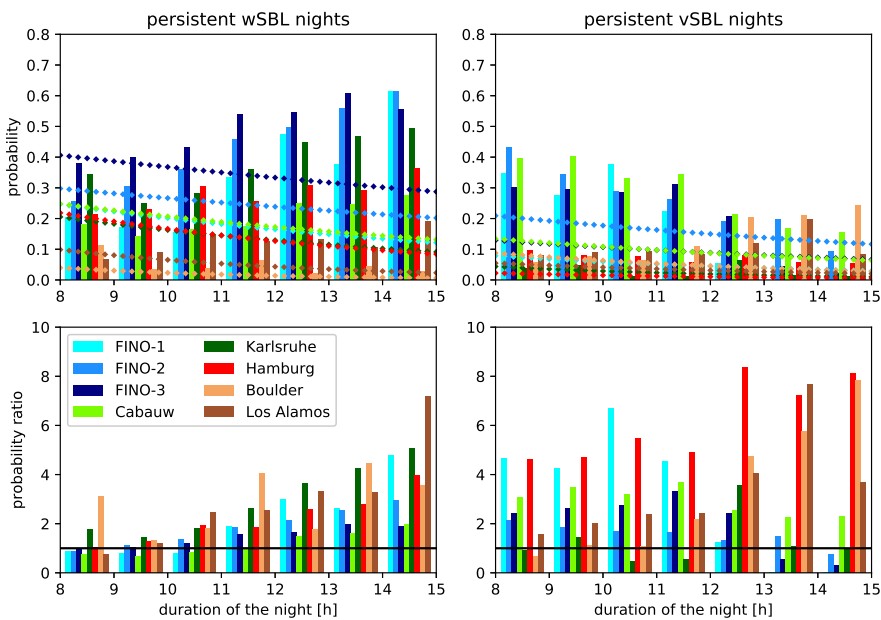

**Figure 11.** Occurrence probabilities of very persistent wSBL (upper left panel, bars) and vSBL (upper right panel, bars) as estimated from the $VP_{ref}$ for nights of different lengths (in one hour increments) at the different tower sites compared to the occurrence probabilities of very persistent nights computed from the FSMC using $\mathbf{Q}_{ref}$ (diamonds). Lower panels show the ratio of the probabilities in the upper panels (values from the $VP_{ref}$ divided by those from the FSMC).

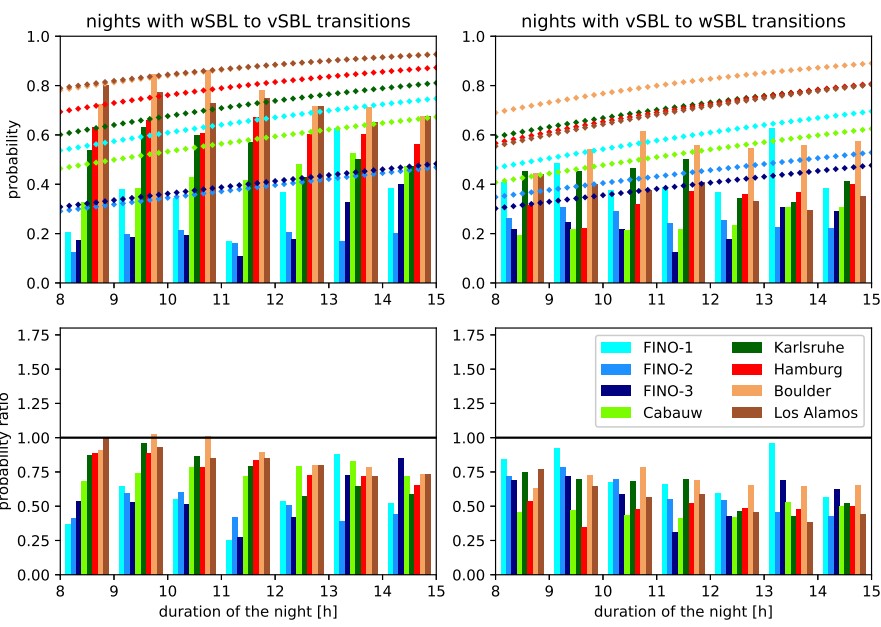

**Figure 12.** As in Figure 11 but for the occurrence probabilities of at least one wSBL to vSBL (left panels) or vSBL and wSBL (right panels) transition within in a night.

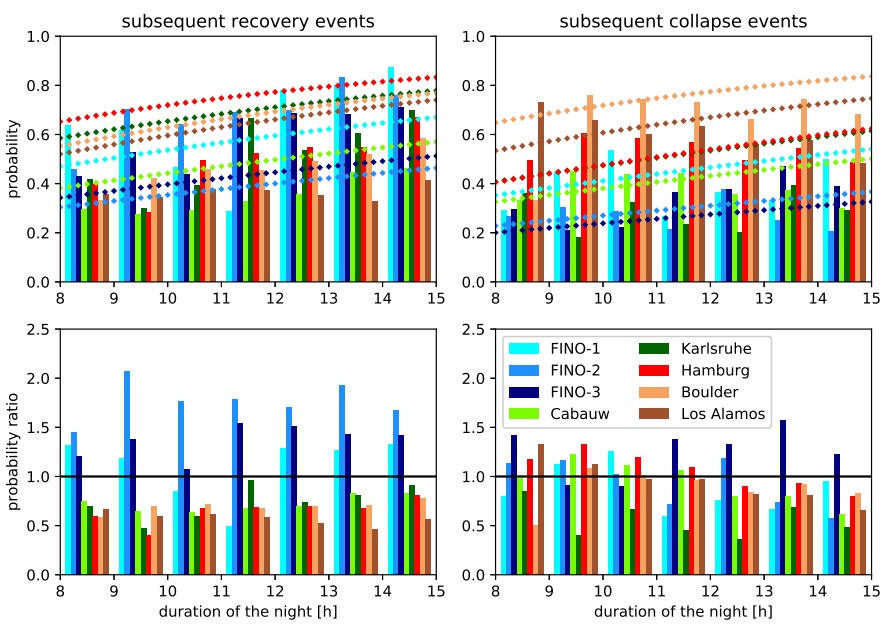

**Figure 13.** As in Figure 11, but for the probabilities of the occurrence of turbulence recovery events subsequent to turbulence collapse (left panels) and turbulence collapse events subsequent to turbulence recovery events (right panels).

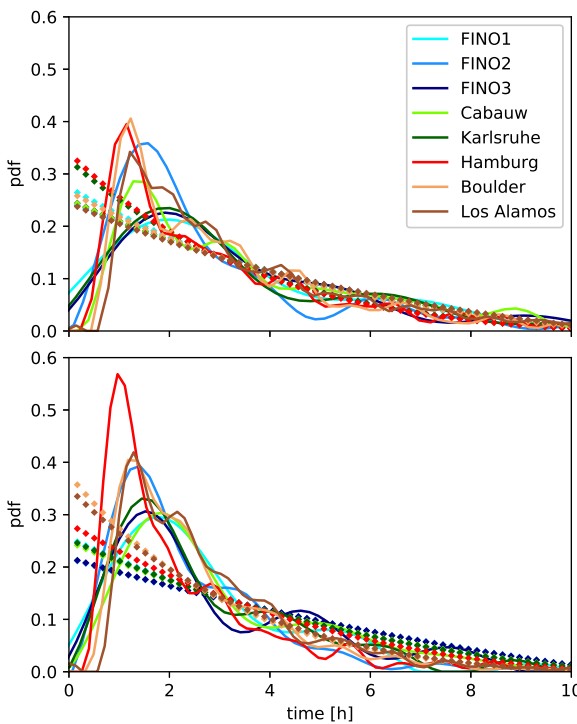

**Figure 14.** Probability density functions of the vSBL (top) and wSBL event durations (bottom) as estimated from the $VP_{ref}$ (solid lines) at the different tower sites compared to FSMC pdfs computed using $\mathbf{Q}_{ref}$ and a nighttime duration of 12 hours (diamonds). All pdfs are calculated with the multivariate kernel density estimation by O'Brien et al. (2014, 2016).

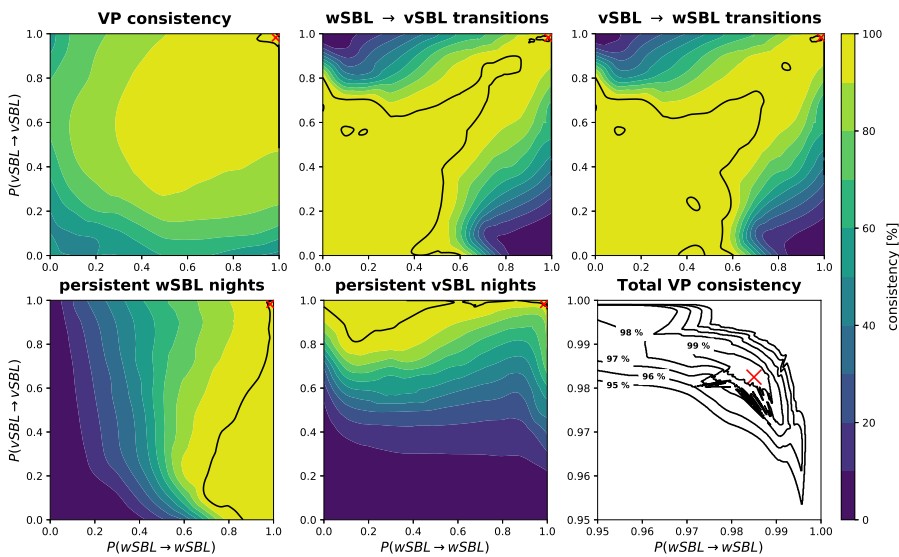

**Figure 15.** Consistency of reference and perturbed regime occupation statistics as functions of Markov chain persistence probabilities. Displayed are: the occupation consistency of the VP (upper left), the consistency of wSBL to vSBL (upper middle) and vSBL to wSBL (upper right) transitions in the VP, the consistency of the occurrence of very persistent wSBL (lower left) and vSBL (lower middle) nights. The 99 % consistency values in each VP characteristic is delineated by a black line. Isolines of the total consistency of the perturbed and reference VP (ranges of persistence probabilities where all SBL regime statistics considered have the same or higher consistencies with $VP_{ref}$) are illustrated in the bottom right panel. In each panel the reference value at Cabauw is shown by a red cross.

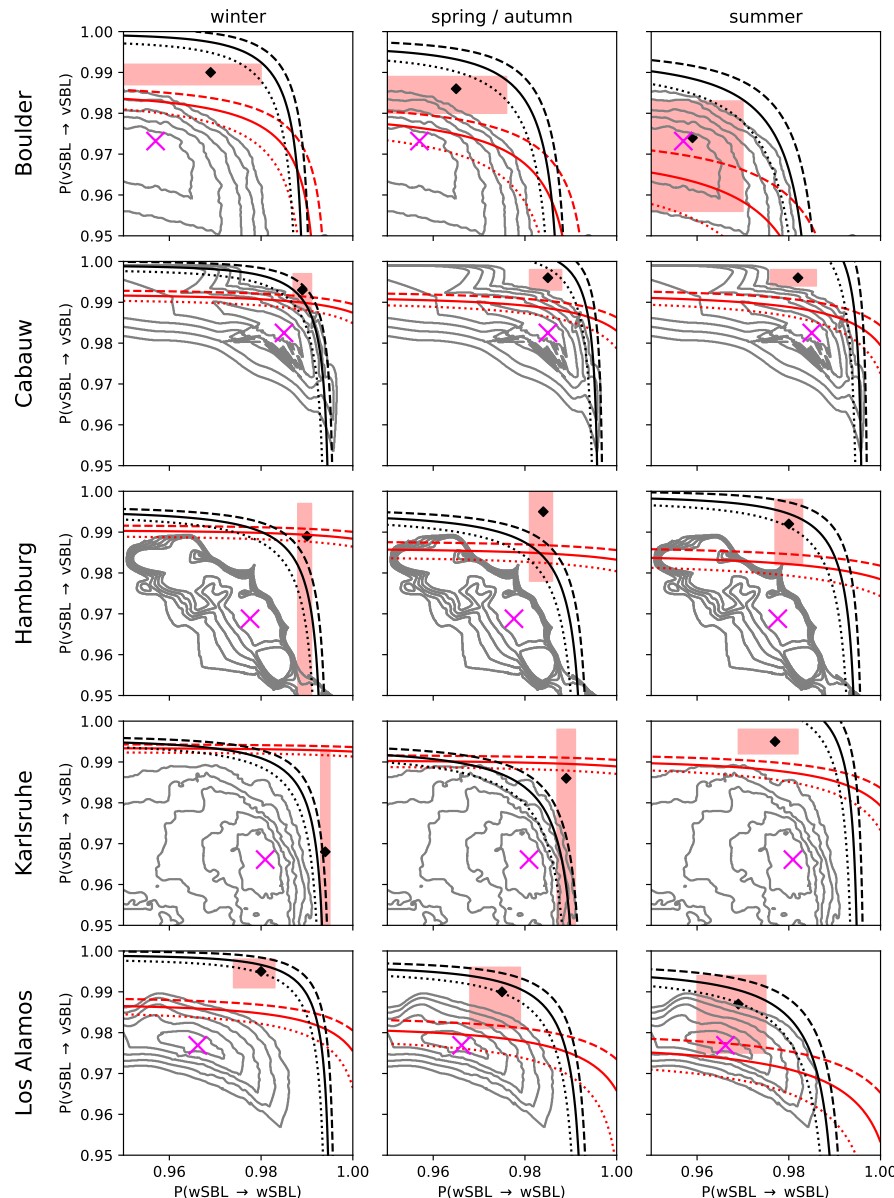

**Figure 16.** Values of persistence probabilities for which the occurrence probability of at least one wSBL to vBSL transition (turbulence collapse) in a night (red lines) or one vSBL to wSBL (turbulence recovery) in a night (black lines) as computed from a stationary Markov chain equal the observed values. Solid, dashed, and dotted lines correspond respectively the observed values, a probability 5 % below the observed values and a probability 5 % above the observed values. The ranges of persistence probabilities where the occurrence probability of very persistent nights in a stationary Markov chain agrees with observations in a $\pm$ 5 % uncertainty band is depicted by the red rectangle with a diamond displaying the values for the exact observational probability occurrence of persistent nights. The persistence probabilities values corresponding to 95 to 99 % total consistency of the perturbed VP with $VP_{ref}$ in the HMM analysis are depicted in grey contours. The persistence probabilities corresponding to $Q_{ref}$ value are marked by a pink cross.

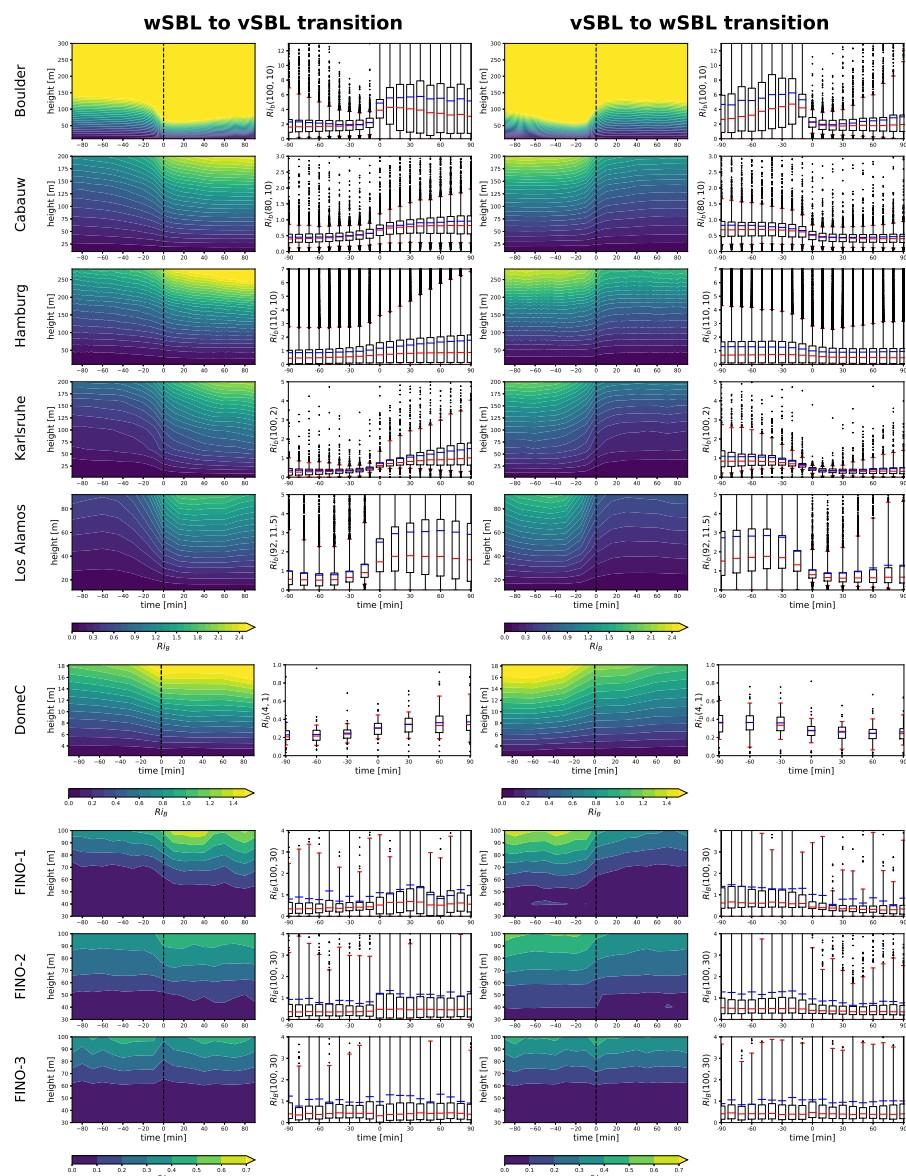

**Figure 17.** Time evolution of the composite median of the bulk Richardson number ($Ri_B$; as determined between each observational level and about 10 m, 1 m, and 30 m for respectively the land-, glacial-, and sea-based tower stations) at the different tower sites in times of turbulence collapse (wSBL to vSBL transition; first and second columns) and turbulence recovery events (vSBL to wSBL transition; third and fourth columns) as determined by the HMM analyses. The composites show the 90 minutes before and after the transitions at time 0 (dashed reference line). Second and fourth row: The distribution of the $Ri_B$ showing the interquartile range (box), 5th to 95th percentile range (outer red bars), median, and mean values (respectively red and blue lines).

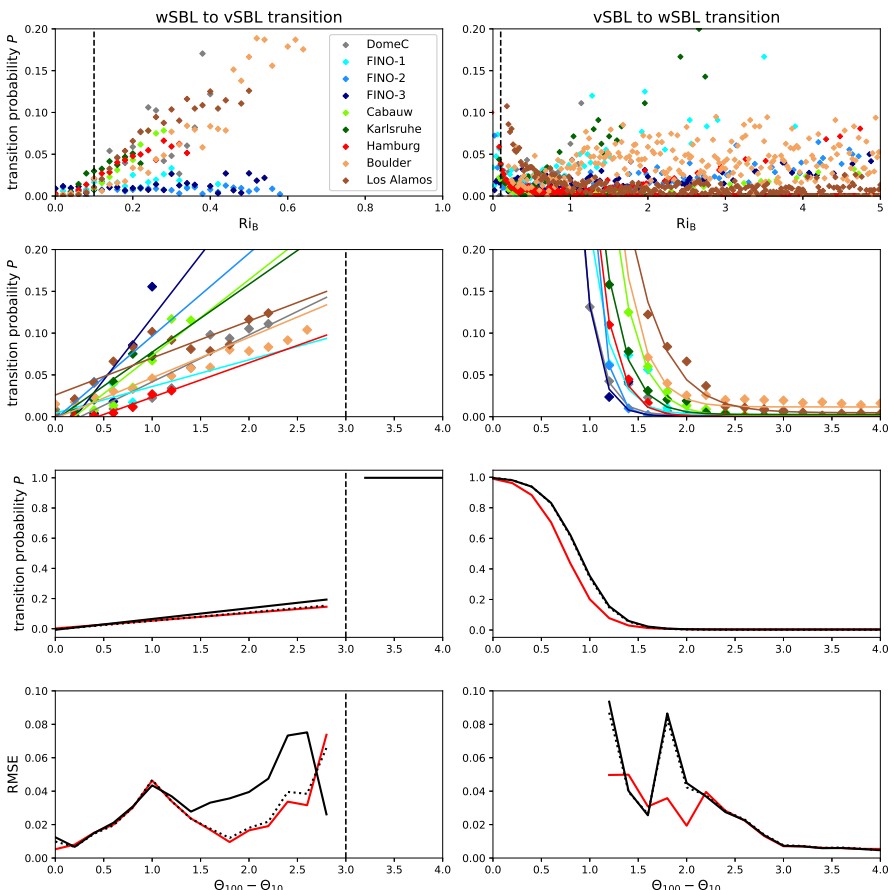

**Figure 18.** First row: Probabilities for wSBL to vSBL (left) and vSBL to wSBL transitions (right) conditioned on the bulk Richardson number (binned by 0.02 increments; coloured diamonds). Second row: Transition probabilities conditioned on the stratification ($\Theta_{100} - \Theta_s$ with the exception $\Theta_4 - \Theta_s$ for DomeC; binned by 0.2 K increments) and best-fit curves. In order to reduce sampling variability in those panels, only data are considered for which the regime occupation probability in a bin exceeds 0.1 % of all data within that regime. Third row: Parameterisation of the state-dependent transition probabilities conditioned on stratification using the mean and median parameter sets of the curve fits (respectively solid and dotted black lines). The best-fit estimated through all stratification data is displayed in red. Fourth row: Root mean square error (RMSE) between the conditional transition probabilities as estimated from HMM anlyses and the parameterised conditional transition probabilities. All transition probabilities have been normalised to 10 minute intervals.

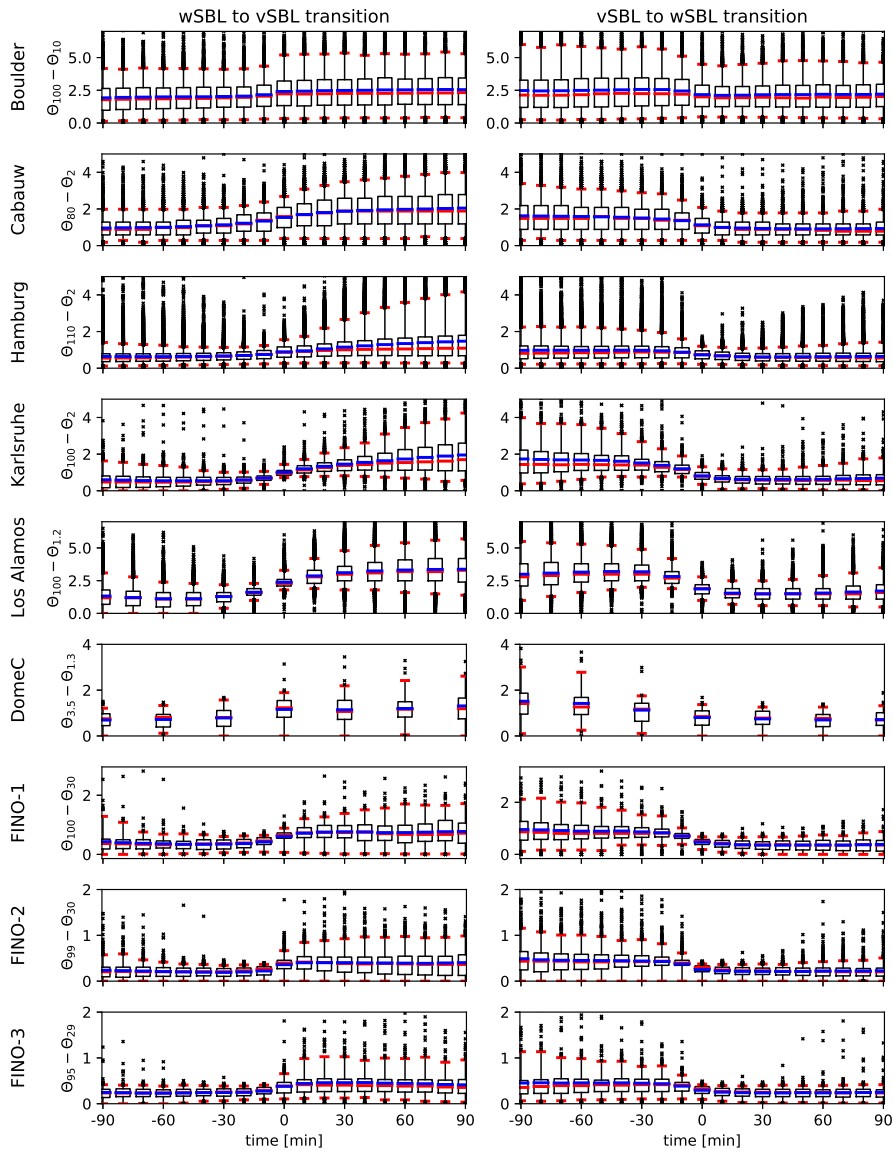

**Figure 19.** Time evolution of the distribution of the stratification as estimated by the potential temperature difference between about 100 m and observations closest to the surface for land-based stations, between about 4 m and 1 m for DomeC, and between about 100 m and 30 m for the sea-based stations in times of wSBL to vSBL (left) and vSBL to wSBL transitions (right). The distributions show the interquartile range (box), 5th to 95th percentile range (outer red bars), median, and mean values (respectively red and blue lines).

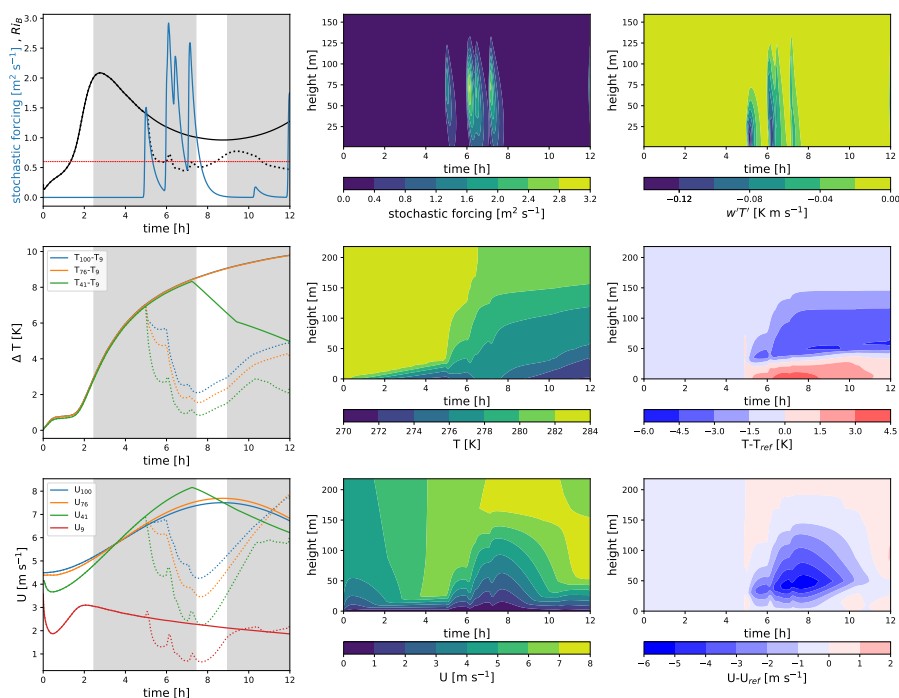

**Figure 110.** One realisation of a twelve hour simulation of the evolution of the nocturnal boundary layer (with time zero being the time the net energy surface flux becomes negative) using the proposed parameterisation. Times when $S$ is in the vSBL are highlighted in grey. Top row from left to right: $Ri_B$ (solid and dotted black lines respectively for the reference experiment without the stochastic parameterisation and the experiment with the stochastic parameterisation) and the strength of the stochastic forcing (blue line; left); the structure of the stochastic forcing function (middle), and the resulting heat fluxes (right). Middle row from left to right: the stratification between different levels (left), the temperature profiles (middle), and the difference in the temperature field to the reference experiment without the stochastic parameterisation (right). Third row from left to right: Wind speeds at different heights (left), wind speed profiles (middle) and the difference in the wind field to the reference experiment without the stochastic parameterisation (right).

**Table 11.** Basic information about the different land-, glacial-, and sea-based tower sites (geographical coordinates, time resolution) and their individual reference HMM state variable inputs $\mathbf{Y}$ (wind speeds $W_h$ and static stabilities $\Delta\Theta$ with their observational levels $h$) and reference transition probability matrices ($\mathbf{Q}_{\text{ref}}$) of HMM analyses estimated from $\mathbf{Y}$. Starting regimes for the transition probability matrices are denoted with a star. Transition probability matrices at Hamburg, Los Alamos, and DomeC are transformed to a 10 minute time resolution, so a direct comparison to other sites is possible. To retrieve original transition probability matrices at these sites the 1/10, 3/2, and 3 matrix powers (respectively) must be taken.

| Tower site | Reference state variables | $\mathbf{Q}_{\text{ref}}$ | | | References |
|---|---|---|---|---|---|
| *Land-based tower sites* | | | | | |
| *Boulder* | $\mathbf{Y} = (W_{100} - W_{10},$ | | *wSBL* | *vSBL* | Kaimal and Gaynor (1983), |
| 40.0500 N, 105.0038 W, 1584 m | $0.5(W_{100} + W_{10}),$ | *wSBL*$^\star$ | 0.9570 | 0.0430 | Blumen (1984) |
| 2008–2015 (10 minute) | $\Theta_{100} - \Theta_{10})$ | *vSBL*$^\star$ | 0.0268 | 0.9732 | |
| *Cabauw* | $\mathbf{Y} = (W_{200} - W_{10}, )$ | | *wSBL* | *vSBL* | Ulden and Wieringa (1996) |
| 51.9700 N, 4.9262 E, -0.7 m | $0.5(W_{200} + W_{10}),$ | *wSBL*$^\star$ | 0.9850 | 0.0150 | |
| 2001–2015 (10 minute) | $\Theta_{200} - \Theta_{2})$ | *vSBL*$^\star$ | 0.0175 | 0.9825 | |
| *Hamburg* | $\mathbf{Y} = (W_{250} - W_{10},$ | | *wSBL* | *vSBL* | Brümmer et al. (2012), |
| 53.5192 N, 10.1051 E, 0.3 m | $0.5(W_{250} + W_{10}),$ | *wSBL*$^\star$ | 0.9776 | 0.0224 | Floors et al. (2014), |
| 2005–2015 (1 minute) | $\Theta_{250} - \Theta_{2})$ | *vSBL*$^\star$ | 0.0312 | 0.9688 | Gryning et al. (2016) |
| *Karlsruhe* | $\mathbf{Y} = (W_{200} - W_{2},$ | | *wSBL* | *vSBL* | Kalthoff and Vogel (1992), |
| 49.0925 N, 8.4258 E, 110.4 m | $0.5(W_{200} + W_{2}),$ | *wSBL*$^\star$ | 0.9809 | 0.0191 | Wenzel et al. (1997), |
| 2003–2013 (10 minute) | $\Theta_{200} - \Theta_{2})$ | *vSBL*$^\star$ | 0.0339 | 0.9661 | Barthlott et al. (2003) |
| *Los Alamos* | $\mathbf{Y} = (W_{92} - W_{11.5},$ | | *wSBL* | *vSBL* | Bowen et al. (2000), |
| 35.8614 N, 106.3196 W, 2263 m | $0.5(W_{92} + W_{11.5}),$ | *wSBL*$^\star$ | 0.9662 | 0.0338 | Rishel et al. (2003) |
| 1995–2015 (15 minute) | $\Theta_{92} - \Theta_{1.2})$ | *vSBL*$^\star$ | 0.0231 | 0.9769 | |
| *Glacial-based tower sites* | | | | | |
| *DomeC* | $\mathbf{Y} = (W_{9} - W_{1.3},$ | | *wSBL* | *vSBL* | Genthon et al. (2010, 2013), |
| 75.1000 S, 123.3000 E, 3233 m | $0.5(W_{9} + W_{1.3}),$ | *wSBL*$^\star$ | 0.9916 | 0.0084 | Vignon et al. (2017a, b) |
| 2011–2016 (30 minute) | $\Theta_{9} - \Theta_{1.3})$ | *vSBL*$^\star$ | 0.0076 | 0.9924 | |
| *Sea-based tower sites* | | | | | |
| *FINO-1* | $\mathbf{Y} = (W_{100} - W_{33},$ | | *wSBL* | *vSBL* | Beeken et al. (2008), |
| 54.0140 N, 6.5876 E, 0 m | $0.5(W_{100} + W_{33}),$ | *wSBL*$^\star$ | 0.9833 | 0.0167 | Fischer et al. (2012) |
| 2004–2015 (10 minute) | $\Theta_{100} - \Theta_{30})$ | *vSBL*$^\star$ | 0.0232 | 0.9768 | |
| *FINO-2* | $\mathbf{Y} = (W_{102} - W_{32},$ | | *wSBL* | *vSBL* | Dörenkämper et al. (2015) |
| 55.0069 N, 13.1542 E, 0 m | $0.5(W_{102} + W_{33}),$ | *wSBL*$^\star$ | 0.9908 | 0.0092 | |
| 2008–2015 (10 minute) | $\Theta_{99} - \Theta_{30})$ | *vSBL*$^\star$ | 0.0138 | 0.9862 | |
| *FINO-3* | $\mathbf{Y} = (W_{100} - W_{30},$ | | *wSBL* | *vSBL* | Fischer et al. (2012) |
| 55.1950 N, 7.1583 E, 0 m | $0.5(W_{100} + W_{30}),$ | *wSBL*$^\star$ | 0.9918 | 0.0082 | |
| 2010–2015 | $\Theta_{95} - \Theta_{29})$ | *vSBL*$^\star$ | 0.0157 | 0.9843 | |

**Table 12.** Nighttime durations ($d$) for the different seasons for estimations of regime statistics from the $VP_{ref}$ and corresponding average durations for calculations in a FSMC.

| Season | $VP_{ref}$ [h] | FSMC [h] |
|---|---|---|
| winter | $13 \leq d$ | 14 |
| spring / autumn | $11 \leq d \leq 13$ | 12 |
| summer | $d \leq 11$ | 10 |

**Table 13.** Probabilities of the occurrence probabilities of at least one wSBL to vSBL or vSBL to wSBL transitions in a night, of the occurrence probabilities of very persistent wSBL or vSBL nights, and of the climatological initial distributions of starting a night in the wSBL or vSBL (respectively $\pi_{\mathrm{wSBL}}$ and $\pi_{\mathrm{vSBL}}$) at the different tower sites for different seasons as estimated from the VP$_{\mathrm{ref}}$.

| Tower station | season | Observations | | | | | |
|---|---|---|---|---|---|---|---|
| | | Transitions | | Very persistent | | clim. | |
| | | wSBL to vSBL [%] | vSBL to wSBL [%] | wSBL nights [%] | vSBL nights [%] | $\pi_{\mathrm{wSBL}}$[%] | $\pi_{\mathrm{vSBL}}$ [%] |
| Land-based stations | | | | | | | |
| Boulder | winter | 68.95 | 56.5 | 3.22 | 22.94 | 45.59 | 54.41 |
| | spring / autumn | 74.84 | 55.18 | 4.44 | 15.43 | 56.24 | 43.76 |
| | summer | 82.07 | 54.41 | 5.32 | 7.29 | 65.2 | 34.8 |
| Cabauw | winter | 48.03 | 31.69 | 29.22 | 14.87 | 73.44 | 26.56 |
| | spring / autumn | 44.75 | 22.37 | 21.36 | 27.68 | 63.16 | 36.84 |
| | summer | 35.99 | 19.1 | 16.49 | 38.92 | 50.50 | 49.50 |
| Hamburg | winter | 54.58 | 38.78 | 37.25 | 5.01 | 87.36 | 12.64 |
| | spring / autumn | 63.16 | 36.26 | 28.36 | 7.02 | 89.77 | 10.23 |
| | summer | 59.94 | 22.86 | 24.89 | 10.81 | 82.22 | 17.78 |
| Karlsruhe | winter | 38.41 | 31.49 | 58.13 | 0.69 | 95.85 | 4.15 |
| | spring / autumn | 49.45 | 41.76 | 40.66 | 3.85 | 89.56 | 10.44 |
| | summer | 40.96 | 22.5 | 13.85 | 32.18 | 57.23 | 42.77 |
| Los Alamos | winter | 65.16 | 30.95 | 13.24 | 16.88 | 74.41 | 25.59 |
| | spring / autumn | 72.99 | 36.92 | 12.86 | 10.13 | 79.46 | 20.54 |
| | summer | 74.32 | 38.57 | 11.73 | 9.58 | 78.75 | 21.25 |
| Ocean-based stations | | | | | | | |
| FINO-1 | winter | 37.84 | 37.84 | 62.16 | 0.00 | 91.89 | 8.11 |
| | spring / autumn | 23.64 | 38.18 | 38.18 | 16.36 | 52.73 | 47.27 |
| | summer | 13.64 | 18.73 | 56.73 | 16.91 | 67.64 | 32.36 |
| FINO-2 | winter | 18.93 | 23.33 | 58.89 | 12.81 | 75.72 | 24.28 |
| | spring / autumn | 18.12 | 24.83 | 47.65 | 22.82 | 64.77 | 35.23 |
| | summer | 15.1 | 26.72 | 28.2 | 40.19 | 39.75 | 60.25 |
| FINO-3 | winter | 31.56 | 29.51 | 57.38 | 6.97 | 86.48 | 13.52 |
| | spring / autumn | 14.08 | 14.79 | 54.23 | 26.06 | 66.2 | 33.80 |
| | summer | 12.23 | 14.61 | 50.32 | 27.16 | 61.36 | 38.64 |

**Table 14.** Parameter values for the state-dependent parametric probability transition functions conditioned on stratification $\Theta_{100} - \Theta_s$ ($\Theta_4 - \Theta_s$ at DomeC) at the different tower locations and the RMSE between parameterised values and those obtained from estimations of HMM analyses. The mean and median values of the parameters are stated below together with the best fit approximation through all data (averaged).

| Tower station | Parameters $P(\text{wSBL} \to \text{vSBL}|\Theta_{100} - \Theta_s)$ | | | Parameters $P(\text{vSBL} \to \text{wSBL}|\Theta_{100} - \Theta_s)$ | | | | |
| --- | --- | --- | --- | --- | --- | --- | --- | --- |
| | $\alpha$ | $\beta$ | RMSE | $\alpha$ | $\beta$ | $\gamma$ | $\delta$ | RMSE |
| Boulder | 0.0484 | -0.0020 | 0.0257 | -0.4953 | 1.060 | 0.4023 | 0.5069 | 0.0061 |
| Cabauw | 0.0909 | -0.0179 | 0.0171 | -0.5012 | 1.0022 | 0.4164 | 0.5023 | 0.0025 |
| DomeC | 0.0562 | -0.0146 | 0.0144 | -0.5009 | 0.7213 | 0.2990 | 0.5027 | 0.0055 |
| FINO-1 | 0.0319 | 0.0042 | 0.0099 | -0.5017 | 0.7736 | 0.3607 | 0.5042 | 0.0116 |
| FINO-2 | 0.0991 | -0.0028 | 0.0067 | -0.5000 | 0.9080 | 0.2140 | 0.5001 | 0.0002 |
| FINO-3 | 0.1495 | -0.0308 | 0.0278 | -0.5001 | 0.7513 | 0.2638 | 0.5007 | 0.0250 |
| Hamburg | 0.0413 | -0.0180 | 0.0084 | -0.5010 | 0.8624 | 0.3284 | 0.5013 | 0.0019 |
| Karlsruhe | 0.0811 | -0.0039 | 0.0084 | -0.5010 | 0.8811 | 0.3896 | 0.5036 | 0.0048 |
| Los Alamos | 0.0443 | 0.0260 | 0.0164 | -0.4985 | 1.0290 | 0.6089 | 0.5032 | 0.0073 |
| mean | 0.0714 | -0.0066 | | -0.5000 | 0.8877 | 0.3648 | 0.5028 | |
| median | 0.0562 | -0.0039 | | -0.5009 | 0.8811 | 0.3607 | 0.5027 | |
| averaged | 0.0513 | 0.0020 | | -0.4997 | 0.7505 | 0.3540 | 0.5045 | |

**Table 15.** Values for the free parameters in the stochastic forcing parameterisation as used in all SCM experiments

| Parameter | symbol | value |
| --- | --- | --- |
| occurrence rate of turbulence pulses | $\lambda_{SF}$ | 5 % per 10 min |
| maximal possible strength of turbulent 'kick' | $R$ | 3 m$^2$ s$^{-1}$ |
| growth time | $\tau_w$ | 600 s |
| eddy overturning timescale | $\tau_e$ | 1200 s |
| centre of turbulent 'kick' at $t_0$ | $h_b$ | 75 m |
| centre of turbulent 'kick' at the end | $h_e$ | 20 m |
| vertical migration timescale of centre | $\tau_h$ | 900 s |
| width of turbulent 'kick' at $t_w$ | $\sigma_w$ | 30 m |
| width of turbulent 'kick' at the end | $\sigma_e$ | 50 m |
| broadening timescale | $\tau_\sigma$ | 900 s |