# Peer review of "A prototype stochastic parameterisation of regime behaviour in the stably stratified atmospheric boundary layer"

_Nonlinear Processes in Geophysics, 2018_

## Referee Comment (RC1) · Anonymous Referee #1 · 14 Nov 2018

Summary: Stationary Hidden Markov models (HMM) are fitted based on long time-series obtained from meteorological tower, using Reynolds averaged meteorological state variables (wind speed, wind shear and stratification). The HMM classifies the data into two regimes, corresponding to weakly stable boundary layers (wSBL) and very stable boundary layers (vSBL). The fitted stationary models are used to obtain statistics of regime occurrences, regime transitions or time between transitions. The HMM estimation provides a transition probability matrix describing regime transitions, and the sensitivity of regime statistics to the matrix values is studied. The authors discuss limitations of the stationarity assumption in the model and acknowledge a need to account for external influences in the transition probability matrix. The dynamics of

transitions are shown not to fit the Markovianity assumption. An idea to use a state dependent Markov model, or regime transition probability matrix, in a turbulent kinetic energy budget closure in a weather or climate model is sketched as a conclusion.

General comments: The idea of including a stochastic representation of SBL regime transitions in a turbulence parameterization for weather or climate models is interesting, and suggesting ways to do so is a welcome contribution. As discussed by the authors in the introduction, models have been proposed to explain transitions from weakly stable to very stable states, but no model exist to represent a recoupling of turbulence to the surface after a decoupled state, or in other words to represent transitions from vSBL to wSBL. Numerous observational studies show events such as gravity waves, instabilities or other types of non-turbulent motions connected to a transition from vSBL to wSBL and such transitions take a rather random character. Therefore, proposing to represent such transitions as a stochastic process is an interesting direction. Yet, the presented study falls short in several aspects. The authors start by discussing HMM analyses of the considered tower data which are presented in parallel papers and which give clear signs of non-stationarity and non-Markovianity in the regime statistics. Nevertheless, the authors choose to present the statistics of regime transitions and occurrences that result from a stationary Markov model and to compare those to the observational statistics, justifying this choice by the wish to test the simplest possible approach. The comparison not surprisingly shows the need to include non-stationarity in the model of regime transitions, as was already discussed by the authors based on the HMM analyses in the cited submitted papers. I am not convinced that this is a very important additional contribution. Discussion of ways to consider non-stationarity in the model is kept to a minimum. Further, the non-stationarity is attributed to external influences, such as synoptic meteorological states (cloud cover, geostrophic wind for example, as was also described in Monahan et al. 2015, JAS). This is a very relevant and important fact, and the work should at the very least discuss methods that provide means of estimating non-stationary models of regime transitions explicitly influenced by external factors, and at best include non-stationarity in the model. Methods to include explicit influence by external factors have been proposed and implemented in atmospheric applications, including to describe SBL regime transitions (eg: Horenko 2010; Metzner et al., 2012; O'Kane et al 2013; Vercauteren and Klein 2015). The need for state dependent transition probabilities, in a stochastic model that would be implemented in the turbulence closure scheme of atmospheric models, is emphasized rightfully and the authors suggest to relate it to the Richardson number. Why not test a Ri number dependence on the transition probability in this paper? That would make the analysis much stronger. The HMM framework is suggested as a foundation for a new parameterization of SBL turbulence, and discussion on how the authors would see such a turbulence parameterization could be expanded. The suggestion is to include random "kicks" of TKE in the vSBL regime, which in turn affect Ri and eventually a transition to wSBL could occur such as no extra TKE source term will be added anymore. Can the authors give ideas on how such a noise term could be defined? And how could such a parameterization fit with the conclusion of the present study, which state: 1- that a stationary Markov chain is inappropriate to represent wSBL to vSBL transition such as driven by radiative cooling. 2- it is inappropriate to represent the statistics of persistent wSBL and vSBL nights as those are impacted by external influences or large-scale synoptic forcing which induces nonstationary behaviour. 3- it could be appropriate to represent a vSBL to wSBL transition after an initial wSBL to vSBL transition. The third point fits with observational and DNS evidence of perturbations that can drive the vSBL back to a wSBL (such as the DNS of Donda et al. 2015, which are cited but not in this context). The authors could also discuss efforts that have been made to describe such "random" perturbations (eg Kang et al. 2014; 2015), which could help giving a stochastic description of the random perturbations, if not of the impact on the TKE itself.

Specific comments:

1- P1 L20: I would suggest to replace "collapsed turbulence" by intermittent turbulence, or turbulence which does obey Monin Obukhov Similarity Theory. More discussion

about modeling difficulties for the turbulence would be appropriate to justify the need for stochastic parameterisation.

2- P2 L30: unrealistic decoupling is also connected to misrepresentation of the TKE. This point could be discussed.

3- P3 L20: The point of representing regime transitions as a stochastic process is clear, but what kind of parameterization is envisioned in each regime? L25: seasonal dependence: is it not more accurately a dependence on external influences? Such influences have been included in non-stationary regime classification schemes (see general comments).

4- P4 L30: the assumptions deserve discussion. The Markovianity assumption could be tested or relaxed, see eg. Franzke et al. 2009. The stationarity assumption is not fulfilled.

5- P5 L15: work on non-stationary statistical clustering should be discussed (see general comments and references).

6- P6 L5: the fact that the influence of seasonal changes is due to changes in the meteorological state means that explicit external influences would improve the model dramatically. Please discuss how to take those into account.

7- Section 4.1: shouldn't the comparison of observation and stationary Markov chain calculation be done for separate time periods? It could be more appropriate to compare the model results with the observational statistics by dividing the dataset in a training part and a control part.

8- L 15-25: Can the results be discussed in light of, eg., the theoretical work of wSBL to vSBL transitions (the MSHF framework by van de Wiel et al. discussed in the introduction). The importance of physical factors highlighted in this model is not included in the Markov model, again potentially calling for inclusion of external factors.

9- L30: here the fact that the stationarity assumption gives satisfactory results is probably consistent with the physical ideas of transitions being linked to random intermittent events. This could be discussed it in the context of existing work (see general comments). Fig 3: Isn't it surprising that transitions are overestimated (or underestimated) for all cases, since the Markov chain is fitted on the data? How can this systematic bias be explained?

10- P7 L5-10 and Fig 5: Do the pdfs show the probability of time spent in a state? The text and the figure caption do not seem to match, or rather, the figure caption is not informative as it is. Moreover I do not really understand the grey band. Why is the width of the distribution so dependent on time after sunset? How is the width of the band calculated? How about the seasonal dependence of the time between transitions? Since it was shown to be critical, why forget it here? Relaxation time: I believe that this could be considered by including finite memory in the Markov model (cf Franzke et al. 2009). The inclusion of explicit external influences should be discussed.

11- P8 L20-30 and Fig 7: each colour has a different number of dots, and the caption does not state what individual dots represent. The discussion actually presents results of an analysis which is different than the one presented in this paper and is already presented in the submitted paper cited as a reference. I do not believe that the results and conclusions should be repeated here. The authors could simply state the conclusions of this parallel study in the discussion.

12- P9 L5: Table 3 only shows the observed probabilities and not a comparison of theoretical and observed.

13- Figure 10: what are the grey dots in the figure?

14- P10 L30: Figure 12 does not exist.

15- P11 L5: how are non-stationarities considered in the analysis? If the stationary Markov chain is defined differently for each season, this is not stated very clearly.

16- P11 L30: "The event duration probability density functions ... display a maximum

an hour or two after sunset" I am confused here. I had understood that the figure showed the pdfs of event duration, or time between two transitions. That has nothing to do with the sunset time (And the sunset time is not mentioned in the figure caption either), but would fit with the recovery time idea which is discussed by the authors.

Horenko I 2010. On the Identification of Nonstationary Factor Models and Their Application to Atmospheric Data Analysis. J. Atmos. Sci. 67:1559–1574.

Metzner P, Putzig L, Horenko I 2012. Analysis of Persistent Nonstationary Time Series and Applications. Commun. Appl. Math. Comput. Sci. 7:175–229.

O'Kane TJ, Risbey JS, Franzke CLE, Horenko I, Monselesan DP 2013. Changes in the Metastability of the Midlatitude Southern Hemisphere Circulation and the Utility of Nonstationary Cluster Analysis and Split-Flow Blocking Indices as Diagnostic Tools. J. Atmos. Sci. 70:824–842.

Kang Y, Belušić D, Smith-Miles K 2014. Detecting and Classifying Events in Noisy Time Series. J. Atmos. Sci. 71:1090–1104.

Kang Y, Belušić D, Smith-Miles K 2015. Classes of structures in the stable atmospheric boundary layer. Q.J.R. Meteorol. Soc. 141:2057–2069.

Franzke CLE, Horenko I, Majda AJ, Klein R 2009. Systematic Metastable Atmospheric Regime Identification in an AGCM. J. Atmos. Sci. 66:1997–2012.

---

## Referee Comment (RC2) · Anonymous Referee #2 · 3 Jan 2019

This is a well-written paper on the characterization of the nocturnal boundary layer in terms of Markov chains. The authors utilize measured data and define a hidden Markov model (HMM). Next they estimate the HMM parameters using a maximum likelihood approach. The work is interesting and the model seems to be well thought. The authors present a convincing comparison of the trained model with observations and they also perform a rigorous sensitivity analysis. The only point that I believe should the authors spend more effort is a more detailed discussion on the justification of the Markov assumption in terms of the physical properties of the nocturnal boundary layer. This should be particularly useful especially for non-experts on SLB.

---

## Referee Comment (RC3) · Anonymous Referee #3 · 25 Jan 2019

The manuscript "Characterising regime behaviour in the stably stratified nocturnal boundary layer on the basis of stationary Markov chains" discusses how well hidden Markov model (HMM) for two distinct regimes (weakly and very stable boundary layers) matches with the observational regime statistic, and how these comparisons help in development of SBL turbulence parameterisations. The authors present a very thorough set of comparisons that lead to detailed conclusions, effectively summarized in the abstract. My understanding is that the stationary HMM is not quite compatible with the observational data, which in itself is a useful conclusion, since it indicates that modified or other types of models are required. Overall, the paper is well-written and the results seem highly relevant, so it could be potentially considered for publication in NPG, except for the point raised in the paragraph below. Other (hopefully) minor comments are listed below as well.

The main problem I had with this paper was that many results and discussions are quite difficult to follow for someone not familiar with AM18a, AM18b, AM18c (which not all available freely on the internet either to refer to - at least I could not locate them), since the authors refer to these three papers quite a lot. It also makes it difficult to assess the novelty of this paper compared to these three papers. (Of course, one natural question is why this paper is not called "Part IV" of the same series and submitted to J.Atmos.Sci. - presumably then the same referees will get to see all the four papers, and better able to judge the quality and novelty of results.)

Other minor comments: (Please note that I have not used the bold-faced symbols $\mathbf{Q}, \mathbf{X}$ etc. in the description below - hopefully it does not cause any confusion.)

(1) Equation (1) should not have $P(x_t = j)$ on the right, and should read:
$$P(x_{t+1} = i | x_t = j, x_{t-1} = k, \ldots, x_0 = n) = Q_{ij}$$

(2) Since equation (2) gives the observational likelihood, conditioned on the states $X$ and the parameters $\lambda$, it should simply involve the product of likelihoods at different times as follows:
$$P(Y|X, \Lambda) = \prod_{t=1}^{T} p(y_t | x_t = i_t, \lambda_{i_t})$$

There is also a problem with the notation in same paragraph: the $K$-dimensional vector of parameters $\lambda_i$ for each $i = 1, 2$ needs to have an additional separate index, e.g., $\lambda_{i,\alpha}$ with $\alpha = 1, \ldots, K$.

The other possibility is that the authors really wanted to write $P(Y, X|\Lambda)$ (without conditioning on the states), in which case the equation is "morally" correct but needs a lot more notational changes:
$$\begin{aligned}
P(Y, X = \{i_1, i_2, \ldots, i_T\}|\Lambda) &= P(Y|X, \Lambda)P(X|\Lambda) \\
&= P(Y|X, \Lambda)P(X) \\
&= \prod_{t=1}^{T} p(y_t | x_t = i_t, \lambda_{i_t}) p(x_1 = i_1) \prod_{t=2}^{T} Q_{i_t, i_{t-1}} \\
&= \pi_{i_1} p(y_1 | x_1 = i_1, \lambda_{i_1}) \prod_{t=2}^{T} p(y_t | x_t = i_t, \lambda_{i_t}) Q_{i_t, i_{t-1}}
\end{aligned}$$

Not pretty, but I just cannot think of any easier way of writing this correctly.

In order to obtain $P(\Lambda|Y)$, of course the above equation needs to be now summed over all possible states, which is a sum over $2^T$ terms (so I am not sure how the authors deal with this problem).

I hope that for the numerical calculations, the authors did indeed use the correct forms of these equations.

The discussion in the appendix also suffers from similar problems with notation, making them difficult to read, in my opinion.

(3) Since the matrix $Q$ in equations (1-2) and $\lambda$ in equation (2) (the "corrected" version) do not have the time index, it is automatically clear that they are time-independent, so comment three is redundant (or another way to put it is that these equations are only valid under the assumption of stationarity). So the authors may want to rephrase that comment.

(4) I am not an expert in the field, but the terms "diel cycle" or "diel nonstationarities" were confusing: even a quick internet search does not bring up anything consistent (the first search result is "Diel vertical migration"!). Are the authors simply using it as a replacement to "diurnal" (which is much more common) or some other technical meaning?

---

## Author Comment (AC2) · 23 Aug 2019

We are very thankful for the reviewer's suggestion to justify the assumptions of the Markov model approach in terms of its applicability for stable boundary layer regime dynamics in more detail. In the introduction of the new manuscript we justify our choice of testing the Markov model assumption as a foundation for SBL regime dynamics (cf. p.4 ll. 1-11). However, as it turns out the Markov model does not suffice to model SBL regime dynamics (p.10 ll. 1-5) we do not overrate its potential and do not try to speculate which particular physical properties might or might not be approximated by a Markov assumption. Instead, we lead the discussion to the finding that additional complexity in a possible stochastic parameterisation is needed (state dependent transition probabilities; section 5.1) and how we can develop a realistic yet simple stochastic parameterisation (cf. section 5.2). Justification for particular choices of building the stochastic parameterisation representing typical physical SBL phenomena are stated in those sections.

The revised manuscript is attached.

Please also note the supplement to this comment:
https://www.nonlin-processes-geophys-discuss.net/npg-2018-44/npg-2018-44-AC2-supplement.pdf
* * *
[Figure]

**Supplement:**

[revised manuscript text omitted]

---

## Author Response (AR1)

Carsten Abraham
School of Earth and Ocean Sciences
Bob Wright Centre A335
University of Victoria
PO Box 1700 STN CSC
Victoria BC V8W 2Y2
Canada

August 23, 2019

Dear Dr. Juan Restrepo,

My co-authors and I are very thankful for the reviewers' comments which resulted in an improved and substantially revised manuscript. In our original version of the manuscript we had investigated the possibility of simulating regime dynamics in the stably stratified nocturnal boundary layer (SBL) with a stochastic parameterisation on the basis of stationary Markov chains. Such an approach is independent of the state of the atmospheric boundary layer. Due to the fact that such a model had not satisfactorily approximate SBL regime dynamics we had envisioned a possible SBL state-dependent explicitly stochastic parameterisation in the discussion. As suggested by reviewer number 1, however, in the revised manuscript we actually have developed this envisioned explicitly stochastic parameterisation. Therefore, we have included a new section to the manuscript in which we do not only develop the parameterisation but also test it in an idealised single column model. The suggested alterations and enhanced analyses have resulted in a substantially restructured and revised manuscript with the following changes:

1. We would like to change the title from "Characterising regime behaviour in the stably stratified nocturnal boundary layer on the vasus of stationary Markov chains" into "A prototype stochastic parameterisation of regime behaviour in the stably stratified atmospheric boundary layer" which is now more appropriate.
2. Amber M. Holdsworth is introduced as a third author.
3. The section about the investigation how well stationary Markov chains can simulate SBL regime dynamics (section 4) has been substantially shortened and summarised to the main key points:
   a. Stationary Markov chains using transition probabilities as estimated from observations are unable to approximate SBL regime dynamics accurately as nonstationarities and non-Markov behaviour are too prevailing.
   b. SBL regime dynamics as estimated from observations are relatively insensitive to the actual transition probability matrix allowing for an investigation of a larger range of different transition probabilities in a stationary Markov chain.
   c. Stationary Markov chains are also unable to approximate SBL regime dynamics of interest even if a relatively broad range of transition probabilities and seasonal nonstationarities are accounted for.

d.  The presented results demonstrate that state-dependent parameterisations for the SBL regime dynamics are necessary.

4.  A fifth section is added evaluating state-dependent transition probabilities conditioned on different stability variables of the SBL. This information is used in order to develop a state-dependent explicitly stochastic parameterisation for SBL regime dynamics which is then tested in an idealised single column model.

5.  The discussion and conclusion section is substantially shortened as the stochastic parameterisation is actually developed and tested. Instead, the potential of such stochastic parameterisation is evaluated.

We think that the revised manuscript is a much improved contribution to the SBL community as it does not only show that stationary Markov chains are inappropriate to simulate SBL regime dynamics but also offers a potential possibility to account for SBL regime dynamics through state-dependent explicitly stochastic parameterisations in models for weather and climate in the near future.

We have adopted the following modifications to the manuscript and replies to the reviewers (**bold**) are provided in *italic* characters. Within the revised manuscript (with tracked changes) modified text is indicated in red.

Thank you very much for considering our manuscript for publication.

Sincerely,

Carsten Abraham

Anonymous Referee #1

Summary: Stationary Hidden Markov models (HMM) are fitted based on long timeseries obtained from meteorological tower, using Reynolds averaged meteorological state variables (wind speed, wind shear and stratification). The HMM classifies the data into two regimes, corresponding to weakly stable boundary layers (wSBL) and very stable boundary layers (vSBL). The fitted stationary models are used to obtain statistics of regime occurrences, regime transitions or time between transitions. The HMM estimation provides a transition probability matrix describing regime transitions, and the sensitivity of regime statistics to the matrix values is studied. The authors discuss limitations of the stationarity assumption in the model and acknowledge a need to account for external influences in the transition probability matrix. The dynamics of transitions are shown not to fit the Markovianity assumption. An idea to use a state dependent Markov model, or regime transition probability matrix, in a turbulent kinetic energy budget closure in a weather or climate model is sketched as a conclusion.

General comments: The idea of including a stochastic representation of SBL regime transitions in a turbulence parameterization for weather or climate models is interesting, and suggesting ways to do so is a welcome contribution. As discussed by the authors in the introduction, models have been proposed to explain transitions from weakly stable to very stable states, but no model exist to represent a recoupling of turbulence to the surface after a decoupled state, or in other words to represent transitions from vSBL to wSBL. Numerous observational studies show events such as gravity waves, instabilities or other types of non-turbulent motions connected to a transition from vSBL to wSBL and such transitions take a rather random character. Therefore, proposing to represent such transitions as a stochastic process is an interesting direction. Yet, the presented study falls short in several aspects. The authors start by discussing HMM analyses of the considered tower data which are presented in parallel papers and which give clear signs of non-stationarity and non-Markovianity in the regime statistics. Nevertheless, the authors choose to present the statistics of regime transitions and occurrences that result from a stationary Markov model and to compare those to the observational statistics, justifying this choice by the wish to test the simplest possible approach. The comparison not surprisingly shows the need to include non-stationarity in the model of regime transitions, as was already discussed by the authors based on the HMM analyses in the cited submitted papers. I am not convinced that this is a very important additional contribution. Discussion of ways to consider non-stationarity in the model is kept to a minimum. Further, the non-stationarity is attributed to external influences, such as synoptic meteorological states (cloud cover, geostrophic wind for example, as was also described in Monahan et al. 2015, JAS). This is a very relevant and important fact, and the work should at the very least discuss methods that provide means of estimating non-stationary models of regime transitions explicitly influenced by external factors, and at best include non-stationarity in the model. Methods to include

explicit influence by external factors have been proposed and implemented in atmospheric applications, including to describe SBL regime transitions (eg: Horenko 2010; Metzner et al., 2012; O'Kane et al 2013; Vercauteren and Klein 2015).

>>>*We are very thankful for the general evaluation of the reviewer and understand very well that our first manuscript came short in many aspects. The manuscript has been substantially revised and restructured to address the reviewer's concerns. As remarked by the reviewer a state-independent HMM-based Markov-chain parameterization (which is investigated to evaluate if such a simple approach suffice to simulate SBL dynamics) is not able to simulate SBL regime dynamics which is why we have substantially shortened and summarised this discussion. Instead, we have extended the discussion by presenting an explicitly stochastic, state-dependent parameterisation of regime dynamics which is able to account for the SBL regime dynamic features of interest.*

The need for state dependent transition probabilities, in a stochastic model that would be implemented in the turbulence closure scheme of atmospheric models, is emphasized rightfully and the authors suggest to relate it to the Richardson number. Why not test a Ri number dependence on the transition probability in this paper? That would make the analysis much stronger. The HMM framework is suggested as a foundation for a new parameterization of SBL turbulence, and discussion on how the authors would see such a turbulence parameterization could be expanded. The suggestion is to include random "kicks" of TKE in the vSBL regime, which in turn affect Ri and eventually a transition to wSBL could occur such as no extra TKE source term will be added anymore. Can the authors give ideas on how such a noise term could be defined? And how could such a parameterization fit with the conclusion of the present study, which state: 1- that a stationary Markov chain is inappropriate to represent wSBL to vSBL transition such as driven by radiative cooling. 2- it is inappropriate to represent the statistics of persistent wSBL and vSBL nights as those are impacted by external influences or large-scale synoptic forcing which induces nonstationary behaviour. 3- it could be appropriate to represent a vSBL to wSBL transition after an initial wSBL to vSBL transition. The third point fits with observational and DNS evidence of perturbations that can drive the vSBL back to a wSBL (such as the DNS of Donda et al. 2015, which are cited but not in this context). The authors could also discuss efforts that have been made to describe such "random" perturbations (eg Kang et al. 2014; 2015), which could help giving a stochastic description of the random perturbations, if not of the impact on the TKE itself.

>>>*We are very thankful for the suggestion of the reviewer to actually write down a state-dependent stochastic parameterisation which is now done in section 5. First we investigate the state-dependent transition probabilities conditioned on internal state variables (cf. section 5.1). In section 5.2 we then develop the complete stochastic parameterisation for first order TKE closure models. During building a prototype stochastic parameterisation, we have decided that if we actually develop the stochastic parameterization, preliminary tests in an idealised single column model would better demonstrate its feasibility. That is the reason why we included some results showing that the conceptual framework shows potential of reasonably well representing the SBL regime dynamics for weather and climate models (cf. section 5.3). Due to the length of the paper (with the new section 5) we refrain from*

*discussing detailed sensitivity analyses of this parameterisation which will be done in a future study.*

**Specific comments:**

**1- P1 L20: I would suggest to replace "collapsed turbulence" by intermittent turbulence, or turbulence which does obey Monin Obukhov Similarity Theory. Discussion paper about modeling difficulties for the turbulence would be appropriate to justify the need for stochastic parameterisation**.
*>>>We have replaced the description of the turbulence in the two different regimes as suggested (cf. p. 1 ll. 18-24). Furthermore, in the revised manuscript we discuss in more detail why current SBL parameterisations of the SBL fail to reproduce the regime dynamics (cf. p. 1 ll. 24-25 to p. 2 ll. 1-7). The need for stochastic parameterisations is described on p. 3 ll. 15-31.*

**2- P2 L30: unrealistic decoupling is also connected to misrepresentation of the TKE. This point could be discussed.**
*>>>The fact that the unrealistic decoupling is related to the misrepresentation of the TKE has been added. (cf. p. 3 l. 4)*

**3- P3 L20: The point of representing regime transitions as a stochastic process is clear, but what kind of parameterization is envisioned in each regime?**
*>>>A better characterisation of stochastic parameterisation, its general idea, and why SBL dynamics might profit from it has been described (cf. p. 4 ll. 3-5).*

**L25: seasonal dependence: is it not more accurately a dependence on external influences? Such influences have been included in non-stationary regime classification schemes (see general comments).**
*>>>As described in the general comments we wanted to first investigate if state-independent parameterisations suffice to simulate SBL regimes. Due to the weakness and inability of stationary Markov chains to account for all SBL regime dynamics of interests, a more complex state-dependent stochastic parameterisation is envisioned. The state-dependent explicitly stochastic parameterisation presented in section 5 should be able to capture such non-stationary behaviour.*

**4-5 P4 L30: the assumptions deserve discussion. The Markovianity assumption could be tested or relaxed, see eg. Franzke et al. 2009. The stationarity assumption is not fulfilled. P5 L15: work on non-stationary statistical clustering should be discussed (see general comments and references).**
*>>>Work on nonstationary approaches to cluster the data are briefly discussed and why we consider the stationary approach in the first place has been justified more clearly (cf. p. 6 ll.11-23).*

**6- P6 L5: the fact that the influence of seasonal changes is due to changes in the meteorological state means that explicit external influences would improve the model dramatically. Please discuss how to take those into account.**

>>>*As described above the state-dependent explicitly stochastic parameterisation should be able to account for the seasonal dependencies through state-dependent transition probabilities.*

**7- Section 4.1: shouldn't the comparison of observation and stationary Markov chain calculation be done for separate time periods? It could be more appropriate to compare the model results with the observational statistics by dividing the dataset in a training part and a control part.**
>>>*We are thankful for the reviewer's reminder to be careful to consider potential overfitting by assessing our model performance against the same data used to estimate model parameters. Our results show that the 'freely-running' stationary Markov chain does generally not suffice to describe SBL regime dynamics as estimated from observations which is why we argue that state-dependent stochastic parameterisations are needed. Therefore, we think that a potential overfitting by using the same data to estimate transition probabilities and regime statistics against which we assess the model performance is not a primary concern.*

**8- L 15-25: Can the results be discussed in light of, eg., the theoretical work of wSBL to vSBL transitions (the MSHF framework by van de Wiel et al. discussed in the introduction). The importance of physical factors highlighted in this model is not included in the Markov model, again potentially calling for inclusion of external factors.**
>>>*The new explicitly stochastic parameterisation is generally able to account for these processes if only in a simplified manner.*

9- **L30: here the fact that the stationarity assumption gives satisfactory results is probably consistent with the physical ideas of transitions being linked to random intermittent events. This could be discussed it in the context of existing work (see general comments).**
>>>*Due to the fact that we have extended the discussion of the new proposed stochastic parameterisation we have shortened the whole Markov chain discussion to a minimum and refrain from discussing how some aspects might or might not be well simulated by Markov chain approximations.*

**10- P7 L5-10 and Fig 5: Do the pdfs show the probability of time spent in a state? The text and the figure caption do not seem to match, or rather, the figure caption is not informative as it is.**
>>>*Yes, the distributions show the time spent in one state or the event duration. Due to the restructuring we have changed the Figure substantially including its caption to make it more informative (cf. Figure 4).*

**Moreover I do not really understand the grey band. Why is the width of the distribution so dependent on time after sunset? How is the width of the band calculated? How about the seasonal dependence of the time between transitions? Since it was shown to be critical, why forget it here?**

*>>>We acknowledge that our description of that analysis was unclear as we compared non seasonal (observations) with seasonal dependencies (FSMC). The event duration pdfs do not change substantially across the seasons except from the fact that the occurrence of longer event durations become more likely as nights simply have more time for such events to occur. Therefore, we analyse all data and compare those to FSMC event duration probabilities computed for nights lasting 12 hours as an intermediate value representative for all seasons. Detailed analyses of the seasons do not add any substantial content to the discussion. We have changed the Figure 4 and have adopted a more consistent description of the results (cf. p. 7 l. 28 to p.8 l.3).*

**Relaxation time: I believe that this could be considered by including finite memory in the Markov model (cf Franzke et al. 2009). The inclusion of explicit external influences should be discussed.**
*>>>The recovery period is now accounted for in our explicitly stochastic parameterisation.*

**11- P8 L20-30 and Fig 7: each colour has a different number of dots, and the caption does not state what individual dots represent. The discussion actually presents results of an analysis which is different than the one presented in this paper and is already presented in the submitted paper cited as a reference. I do not believe that the results and conclusions should be repeated here. The authors could simply state the conclusions of this parallel study in the discussion.**
*>>>As suggested we have removed this Figure and the discussion from paper.*

**12- P9 L5: Table 3 only shows the observed probabilities and not a comparison of theoretical and observed.**
*>>>That is correct. We have changed the formulation that it is clearer that Table 3 shows only the occupation statistics as estimated from observations and we compare those to calculations in the freely-running Markov chain as presented in the Figures. (cf. p. 9 ll. 9-11).*

**13- Figure 10: what are the grey dots in the figure?**
*>>>The grey dots are regions where the total VP consistency frequently changes. This Figure, however, has been removed from the discussion as it did not add any important information for the discussion and the results have been briefly summarised (cf. p. 9 ll. 28-33).*

**14- P10 L30: Figure 12 does not exist.**
*>>> The enumeration of the figures in the submitted manuscript is such that "Figure 12" denotes the second figure.*

**15- P11 L5: how are non-stationarities considered in the analysis? If the stationary Markov chain is defined differently for each season, this is not stated very clearly.**
*>>>Due to the restructuring this part has been eliminated from the discussion.*

**16- P11 L30: "The event duration probability density functions . . . display a maximum an hour or two after sunset" I am confused here. I had understood that the figure showed the pdfs of event duration, or time between two transitions. That has nothing**

**to do with the sunset time (And the sunset time is not mentioned in the figure caption either), but would fit with the recovery time idea which is discussed by the authors.**
>>>*We are very thankful to the reviewer for catching this mistake. This was completely wrong and the sentences have been eliminated.*

**Anonymous Referee #2**

**This is a well-written paper on the characterization of the nocturnal boundary layer in terms of Markov chains. The authors utilize measured data and define a hidden Markov model (HMM). Next they estimate the HMM parameters using a maximum likelihood approach. The work is interesting and the model seems to be well thought. The authors present a convincing comparison of the trained model with observations and they also perform a rigorous sensitivity analysis. The only point that I believe should the authors spend more effort is a more detailed discussion on the justification of the Markov assumption in terms of the physical properties of the nocturnal boundary layer. This should be particularly useful especially for non-experts on SLB.**

>>>*We are very thankful for the reviewer's suggestion to justify the assumptions of the Markov model approach in terms of its applicability for stable boundary layer regime dynamics in more detail. In the introduction of the new manuscript we justify our choice of testing the Markov model assumption as a foundation for SBL regime dynamics (cf. p.4 ll. 1-11). However, as it turns out the Markov model does not suffice to model SBL regime dynamics (p.10 ll. 1-5) we do not overrate its potential and do not try to speculate which particular physical properties might or might not be approximated by a Markov assumption. Instead, we lead the discussion to the finding that additional complexity in a possible stochastic parameterisation is needed (state dependent transition probabilities; section 5.1) and how we can develop a realistic yet simple stochastic parameterisation (cf. section 5.2). Justification for particular choices of building the stochastic parameterisation representing typical physical SBL phenomena are stated in those sections.*

**Anonymous Referee #3**

>>>*We are very thankful to the reviewer for the suggestion to improve our manuscript, in particular for the detailed correction of our equations describing the hidden Markov model. We did not submit the current study as a Part IV extension to our J.Atmos.Sci. series as that paper series deals with determining the climatology of SBL regime dynamics across the tower data which is also used for this study. In contrast to those climatologies, the paper at hand is intended to work towards stochastic parameterisations of turbulence in the SBL which we found more appropriate as a stand alone paper in this Journal. As we have included the discussion of a parameterisation and its test in a single column model (as suggested by reviewer#1) which has lead to a complete new paper structure, the paper is related to the previous studies to a lesser extent.*

*Comments 1-3: As mentioned above we are very thankful for the careful and detailed corrections of our equations describing the assumptions for the HMM analysis. We have*

*adopted your suggestions to correct our equations and hope that they have also become easy to understand (cf. section 3 of the new manuscript).*

*Comment 4: The term "diel" has been changed to "diurnal"*

[revised manuscript text omitted]

---

## Author Response (AR2)

Carsten Abraham
School of Earth and Ocean Sciences
Bob Wright Centre A335
University of Victoria
PO Box 1700 STN CSC
Victoria BC V8W 2Y2
Canada

October 8, 2019

Dear Dr. Juan Restrepo,

My coauthors and I are very thankful for the comments of the reviewers and your editorial decision of considering our manuscript for publication. We have revised the manuscript carefully to catch typographic errors and improve the readability of the text. As suggested, the labels and tick marks in Figures have been increased.

Sincerely,

Carsten Abraham

[revised manuscript text omitted]
(\mathrm{wSBL} \to \mathrm{vSBL}|\mathrm{Ri_B})$ estimated from using the $\mathrm{VP_{ref}}$ (binned by $\mathrm{Ri_B}$ increments of 0.02) shows low transition probabilities across all tower sites (well below 0.01) for $\mathrm{Ri_B}$ smaller than about 0.1 (Figure 18, upper left panel). For $\mathrm{Ri_B}$ larger than 0.1, $P(\mathrm{wSBL} \to \mathrm{vSBL}|\mathrm{Ri_B})$ increases linearly at the land-based and glacial-based  stations to $\mathrm{Ri_B} \simeq 0.6$ beyond which wSBL conditions are unsustainable. Consistent with the composites in Figure 17, $P(\mathrm{wSBL} \to \mathrm{vSBL}|\mathrm{Ri_B})$ at sea-based stations is independent of $\mathrm{Ri_B}$.

At land-based stations, $P(\mathrm{vSBL} \to \mathrm{wSBL}|\mathrm{Ri_B})$ demonstrates that vSBL conditions below $\mathrm{
[revised manuscript text omitted]
\left((\text{wSBL} \to \overbrace{\text{vSBL} \to \ldots \to \text{vSBL}}^{t\times} \to \text{wSBL}|n) > 0\right) = \sum_{t_1=0}^{n-t-2} (\pi^T \mathbf{Q}^{t_1})_{\text{wSBL}}$$

$$P(\text{wSBL} \to \text{vSBL})P(\text{vSBL} \to \text{vSBL})^t P(\text{vSBL} \to \text{wSBL})\left[ P(\text{wSBL} \to \text{wSBL})^{n-t-t_1-2} \right. \tag{A5}$$

$$\left. + \sum_{t_2=0}^{n-t-t_1-3} P(\text{wSBL} \to \text{wSBL})^{t_2} P(\text{wSBL} \to \text{vSBL})P(\text{vSBL} \to \text{vSBL})^{n-t-t_1-t_2-3} \right],$$

where $\pi$ is the vector of climatological initial probabilities.

5   To calculate the overall probability that such a subsequent event occurs is then the summation over all possible $t$:

$$\sum_t Pr\left((\text{wSBL} \to \overbrace{\text{vSBL} \to \ldots \to \text{vSBL}}^{t\times} \to \text{wSBL}|n) > 0\right) = \
[revised manuscript text omitted]